# Within-species floral evolution reveals convergence in adaptive walks during incipient pollinator shift

Katherine E. Wenzell [1,2,3,4,5], Mikhaela Neequaye [1,5], Pirita Paajanen [1], Lionel Hill[1], Paul Brett[1] & Kelsey J. R. P. Byers [1] ✉

Understanding how evolution proceeds from molecules to organisms to interactions requires integrative studies spanning biological levels. Linking phenotypes with associated genes and fitness illuminates how adaptive walks move organisms between fitness peaks. Floral evolution can confer rapid reproductive isolation, often converging in association with pollinator guilds. Within the monkeyflowers (*Mimulus* sect. *Erythranthe*), yellow flowers within red hummingbird-pollinated species have arisen at least twice, suggesting possible pollinator shifts. We compare two yellow-flowered forms of *M. cardinalis* and *M. verbenaceus* to their red counterparts in floral phenotypes, biochemistry, transcriptomic and genomic variation, and pollinator interactions. We find convergence in ongoing adaptive walks of both yellow morphs, with consistent changes in traits of large effect (floral pigments, associated gene expression), resulting in strong preference for yellow flowers by bumblebees. Shifts in scent emission and floral opening size also favor bee adaptation, suggesting smaller-effect steps from hummingbird to bee pollination. By examining intraspecific, incipient pollinator shifts in two related species, we elucidate adaptive walks at early stages, revealing how convergent large effect mutations (floral color) may drive pollinator attraction, followed by smaller effect changes for mechanical fit and reward access. Thus, ongoing adaptive walks may impact reproductive isolation and incipient speciation via convergent evolution.

Understanding how adaptations in complex traits arise remains a central question in evolutionary biology[1,2]. Theory predicts that adaptive walks to new fitness optima involve a few mutations of large effect and many steps of smaller-effect changes[3], but uncertainty remains regarding how adaptive walks proceed at intermediate stages and when moving between peaks in a fitness landscape[4]. Linking phenotype to genotype to fitness within multilevel systems in an integrative way[5] allows us to assess the repeatability and convergence of adaptation. One such multilevel system is floral trait evolution, which can display phenotypic convergence via selection by specific pollinator guilds, which may also confer reproductive isolation[6,7]. Pollinators are important for flowering plant diversification[6,8–10], but questions remain about how pollinator-mediated speciation arises at early stages from intraspecific floral variation[11–13]. Combining knowledge of key genes underlying floral shifts and their impact on pollinator behavior thus provides a powerful assessment of how evolution drives species diversity.

[1]John Innes Centre, Norwich, UK. [2]Department of Natural Resource Management, South Dakota State University, Brookings, SD, USA. [3]California Academy of Sciences, San Francisco, CA, USA. [4]University of Maryland, College Park, MD, USA. [5]These authors contributed equally: Katherine E. Wenzell, Mikhaela Neequaye. ✉e-mail: Kelsey.Byers@jic.ac.uk

Previous work suggests that genes of large effect can alter floral traits and therefore reproductive isolation via pollinator shifts[14–16], which often involves floral color transitions[17–20]. Understanding the underlying genetic changes can reveal evolutionary mechanisms driving phenotypic changes, e.g. drift or selection[21,22], which may result in phenotypic convergence via independent mutations[23–26], sorting of standing ancestral variation[27,28], or adaptive introgression via gene flow[29,30]. Nonetheless, few if any studies have assessed genetic changes associated with repeated natural floral color transitions in multiple related species and experimentally quantified their impact on pollinator behavior, an approach that would shed light on the repeatability and convergence of pollinator-mediated evolution.

Whether recent floral color transitions are accompanied by changes in other floral traits impacting pollinator behavior remains uncertain[31,32] and provides an opportunity to explore how adaptive walks may proceed between peaks in a fitness landscape. Because suites of floral traits adapted to different pollinator guilds represent a key example of complex trait adaptation[20], exploring variation in additional floral traits in intraspecific floral color transitions (hypothesized to represent pollinator shifts[33], provides insight into the nature of early stages of adaptive evolution. Within the genus *Mimulus* (monkeyflowers, Phrymaceae, *sensu*[34], section *Erythranthe* offers ideal case studies to investigate floral trait evolution via pollinator-mediated selection[14,35,36] and has become an emerging model system for floral trait evolution[37–39]. Section *Erythranthe* includes two independent shifts from insect to hummingbird pollination[40,41]. These shifts include transitions to red flowers, which result from high concentrations of anthocyanins and carotenoids[42,43], in addition to changes in floral morphology, scent, and reward[35,44].

Within the two geographically widespread hummingbird-pollinated members of section *Erythranthe*, *M. cardinalis* and *M. verbenaceus*, naturally-occurring yellow-flowered morphs have arisen and have become well-established in several populations at range edges[33]. Previous work on these yellow morphs found evidence that they were preferred by bumblebee pollinators compared to the red-flowered wild type in both species[45], as well as evidence of assortative mating within the yellow morph due to pollinator fidelity[33]. Yellow floral color was hypothesized to result from a loss of anthocyanins via a single recessive locus[33]. In *M. cardinalis* this is due to expression changes or deletion of the gene *PELAN*, which is responsible for the production of anthocyanins in petal lobes[42]. Despite previous work on these color polymorphisms, it remains unknown whether these stable floral morphs also differ in other pollinator-relevant floral traits (including floral morphology, nectar, pigments, and floral scent), which may represent additional steps along an adaptive walk between hummingbird and bee pollination syndromes by further adapting these morphs to bee pollination via modifications to floral attraction and mechanical fit traits. Does the evolution of novel color morphs with differential pollinator attraction involve only color shifts (i.e., first step mutations of large effect), or are they also altering in more complex ways (via many traits of small effect) to adapt to a novel pollinator?

In order to identify whether these repeated floral color transitions may represent early stages of pollinator-mediated evolution, we investigated whether these independent shifts showed evidence of convergent evolution between the two species (*sensu*[46]) at the phenotypic and genetic levels. We address the following: 1) What are the phenotypic, biochemical, and genetic changes underlying the red-yellow shifts? 2) Is there evidence for convergence between the yellow morphs of both species in floral color and other traits? 3) Are shifts across floral traits of yellow morphs consistent with an adaptive walk toward bee pollination? This approach assesses if phenotypic and genetic changes are best explained by shared (i.e., adaptive introgression of alleles or a shared ancestral polymorphism) or independent (i.e., convergence due to consistent selection pressure by pollinators) sources of genetic variation. By expanding our understanding of the direction, order, and integration of the evolution of floral traits in response to an incipient pollinator shift, this work sheds light on how adaptive walks may proceed at intermediate stages, as complex phenotypes move between peaks in a fitness landscape. Integrating studies of floral phenotypic, biochemical, and genetic variation with their impact on pollinator behavior within multiple species provides a comprehensive investigation of the repeatability of early stages of adaptive evolution.

## Results

### Abbreviations and summary
We used two existing inbred lines of *M. verbenaceus* (red: MVBL, hereafter MvR; yellow: MVYL, hereafter MvY, see Methods) and two existing inbred lines of *M. cardinalis* (red: CE10, hereafter McR; yellow: SM, hereafter McY).

To characterize these repeated floral transitions from red to yellow flowers, we compared MvY and McY to their red conspecific forms in terms of floral phenotypes, biochemistry, transcriptomic and genomic variation, and interactions with bumblebee pollinators. Despite contrasting differences in some aspects of floral phenotype (nectar guides, corolla tube length, and scent composition), we find evidence of convergent evolution in pigments, gene expression, and pollinator responses in both yellow forms (Table 1). Yellow morphs share less restrictive floral tubes, increased scent emission, and patterns of carotenoid production and expression of key carotenoid biosynthetic genes. Further, we find some evidence that homologues of R2R3-MYB transcription factor (*PELAN*) are likely involved in the lack of anthocyanin expression in the petal lobes of both yellow forms, which attract novel bumblebee pollinators at a rate of 2:1 over red flowers. Finally, analysis of whole genome sequence variation reveals independent paths of molecular evolution, even when the same genes are involved, consistent with convergent evolution of yellow flowers with increased attractiveness to bee pollinators.

### Floral color: steps of large effect in an adaptive walk
We compared spectral reflectance of petal lobes of all morphs by plotting reflectance curves and performing principal component analysis of reflectance values at wavelengths from 300 to 700 nm. We found that red flowers of both species (MvR and McR) appeared similar in both human vision and reflectance curves, while yellow morphs (MvY and McY) differed in both human vision and reflectance (Fig. 1A, B; Supplementary Fig. S2). While petal lobes of both red morphs reflect almost no UV, MvY and particularly McY are modestly reflective in the UV range, with a subtle bullseye pattern in both (Supplementary Fig. S3).

Total anthocyanins extracted from corolla tissues did not vary among species ($F_{1, 16} = 0.89$, $p = 0.36$; Fig. 1C) but varied between color morphs within species ($F_{2,16} = 207.8$, $p < 0.0001$), with higher levels in red morphs than their yellow conspecifics (MvR-MvY: $p < 0.0001$; McR-McY: $p < 0.001$; Supplementary Table S2). Total carotenoids varied significantly among species ($F_{1, 16} = 15.2$, $p = 0.001$; Fig. 1D) and between color morphs within species ($F_{2,16} = 14.04$, $p = 0.0003$). Both yellow-flowered morphs had significantly higher total carotenoids than their red conspecifics (MvR-MvY: $p = 0.0028$; McR-McY: $p = 0.032$; Supplementary Table S2). This suggests that both species' shifts from red to yellow flowers were mediated by a simultaneous loss of anthocyanins and an increase in carotenoids.

### Individual anthocyanins
We characterized the individual anthocyanins produced by each morph and found that anthocyanin accumulation did not show any consistent patterns across morphs (Fig. 2A). Two core anthocyanidins (cyanidin and pelargonidin) and eight anthocyanins were found in the two species. Three anthocyanins were found in more than one morph: cyanidin-glucoside, cyanidin-glucoside-rhamnose and

**Table 1 | Summarized differences in yellow (MvY and McY) vs red morphs (MvR and McR, respectively) spanning floral phenotypes, gene expression, and pollinator response**

| | *M. verbenaceus* MvY (vs MvR) | *M. cardinalis* McY (vs McR) |
|---|---|---|
| **Anthocyanin pigments Floral color** | | |
| Anthocyanins (total) | – | – |
| Cyanidin-Glucoside | Absent | – |
| Pelargonidin-Glucoside | Absent | Absent in both |
| Cyanidin-Glucoside-Rhamnose | + (absent in MvR) | – |
| Pelargonidin-Glucoside-Rhamnose | + (absent in MvR) | – |
| Pelargonidin-Pentose | Absent | Absent in both |
| Isomer of Pelargonidin-Glucoside-Rhamnose | Absent | Absent in McY |
| Cyanidin-Glucoside-Acetate | Absent | Absent in both |
| Pelargonidin-Glucoside-Acetate | Absent | Absent in both |
| **Anthocyanin regulatory gene expression** | | |
| *PELAN* | + | – (deleted) |
| *NEGAN* | Absent in both | + |
| **Carotenoid pigments** | | |
| Carotenoids (total) | + | + |
| Zeaxanthin | n.c. | – |
| Antheraxanthin | + | n.c. |
| Violaxanthin | + | + |
| Neoxanthin | + | + |
| Mimulaxanthin | + | + |
| **Carotenoid biosynthesis gene expression** | | |
| *BCH* | + | + |
| *ZEP* | + | + |
| **Floral scent** | | |
| Total volatile emissions | + | + |
| **Floral morphology** | | |
| Corolla length | + | – |
| Opening width, height | +, + | +, + |
| Display width, height | n.c., + | n.c., n.c. |
| **Nectar** | | |
| Nectar volume | n.c. | n.c. |
| Sugar concentration | + | n.c. |
| **Bumblebee preference** | | |
| Flowers visited, probed | +, n.c. | +, + |
| Handling time | + | + |

Plus sign (+) indicates an increase in yellow morphs compared to red; minus (–) indicates a decrease; n.c. denotes no change. See text for details. Trait categories/headings in bold.

pelargonidin-glucoside-rhamnose. The lowest total anthocyanin accumulation was found in the wholly yellow MvY, which produced cyanidin-rhamnose-glucoside and pelargonidin-rhamnose-glucoside at almost undetectable levels. In contrast, McY, which has red nectar guides (Fig. 1A), produces both anthocyanins found in MvY as well as cyanidin-glucoside. Of these three yellow morph anthocyanins, cyanidin-glucoside-rhamnose was found at the highest levels in McY, with the others at barely detectable levels. In contrast to McY, McR accumulates 2.4-fold more cyanidin-glucoside-rhamnose (Welch's $t$ test, $t = 8.899$, df = 4, $p = 0.0009$) in addition to 15.6-fold more pelargonidin-glucoside-rhamnose (Welch's $t$ test, $t = 9.763$, df = 4, $p = 0.0006$). In addition to the three anthocyanins shared with McY,

McR also produces an additional isomer of pelargonidin-glucoside-rhamnose. In contrast with the other morphs, MvR shows the highest diversity of anthocyanins (five, three unique to MvR). Interestingly, MvR does not share a single anthocyanin with MvY. This is in contrast with McY, which shares all three of its anthocyanins with McR. Moreover, MvR is the only line to accumulate anthocyanins with pentose and acetate decorations. In addition, pelargonidin-glucoside is only found in MvR, whereas pelargonidin-glucoside-rhamnose is found in the other three morphs. This suggests a potential loss of ability to add rhamnose in addition to the gain of ability to add pentose and acetate decorations in MvR. MvR also contains significantly higher levels of cyanidin-glucoside when compared to the two *M. cardinalis* lines (Welch's $t$ test, $t = 13.14$ df = 4, $p = 0.0002$ for McR, $t = 13.99$ df = 4, $p = 0.0002$ for McY).

### Anthocyanin biosynthesis and regulatory genes

To better characterize the loss of red pigmentation in the yellow morphs, we analyzed expression of anthocyanin biosynthesis pathway genes in corolla tissue. No significant difference was found between core anthocyanin biosynthesis genes across the four morphs, with the exception of *chalcone synthase* (*CHS*) and *flavanone-3-hydroxylase* (*F3H*, only significantly different in *M. verbenaceus*) (Fig. 2B). *CHS* is significantly upregulated in both yellow morphs compared to their red counterparts (Welch's $t$ test, $t = 13.02$ df = 2, $p = 0.0058$ for McY, $t = 8.880$ df = 2, $p = 0.0124$ for MvY). These enzymatic reactions occur early in anthocyanin biosynthesis and lead not only to the production of anthocyanins but also other flavonoids (Supplementary Table S6)[47,48].

By contrast, significant differences were found in previously characterized *Mimulus* anthocyanin regulatory genes[42]. *NEGAN* is only expressed in McY (Fig. 2C), consistent with published work demonstrating its role in McY's red-spotted corolla throat[42]. MvY does not have any red spots or associated *NEGAN* expression. The published *PELAN* is highly upregulated in MvY despite accumulating the lowest total anthocyanins of the four morphs (Welch's $t$ test, $t = 6.117$ df = 2, $p = 0.0257$). Closer inspection of the whole genomic sequence in order to confirm the identity of this *PELAN* homolog showed it is deleted in McY, again in agreement with[42] – whereas it is both present and highly expressed in MvY. Six *PELAN* homologs were found in the two species' genomes, all clustered on chromosome 4. These were named *PELAN*(1-6) with *PELAN*(1) being the published *PELAN*. *PELAN*(3) and *PELAN*(4) were not expressed in corolla tissue of any line. *PELAN*(6) was only expressed in MvR and *PELAN*(2) and *PELAN*(5) were only expressed in McR (Fig. 2C, Supplementary Table S7). This suggests a historical set of duplications resulting in multiple diverged copies of *PELAN* with various functions or expression domains, which likely explains the lack of consistency found in the different anthocyanins in these morphs.

### Flavonoids

Due to the increased expression of *CHS* (the enzyme responsible for the first committed step in flavonoid biosynthesis, 47) in yellow morphs, we also briefly investigated the accumulation of additional flavonoids. Preliminary analyses could not find a relationship between anthocyanins, their core biosynthesis genes, and flavonoids detectable at a wavelength of 350 nm (Supplementary Fig. S12, Supplementary Table S8).

### Individual carotenoids

Yellow morphs had higher levels of total carotenoids compared to their red conspecifics. In order to better characterize this shift, we quantified individual carotenoids. While none of the eight anthocyanins characterized could be found across all morphs, several carotenoids – first characterized in *Mimulus* by[43] – were detected across all four lines (Fig. 3A, Supplementary Table S9). Zeaxanthin (first in the biosynthesis pathway) is found in significantly higher

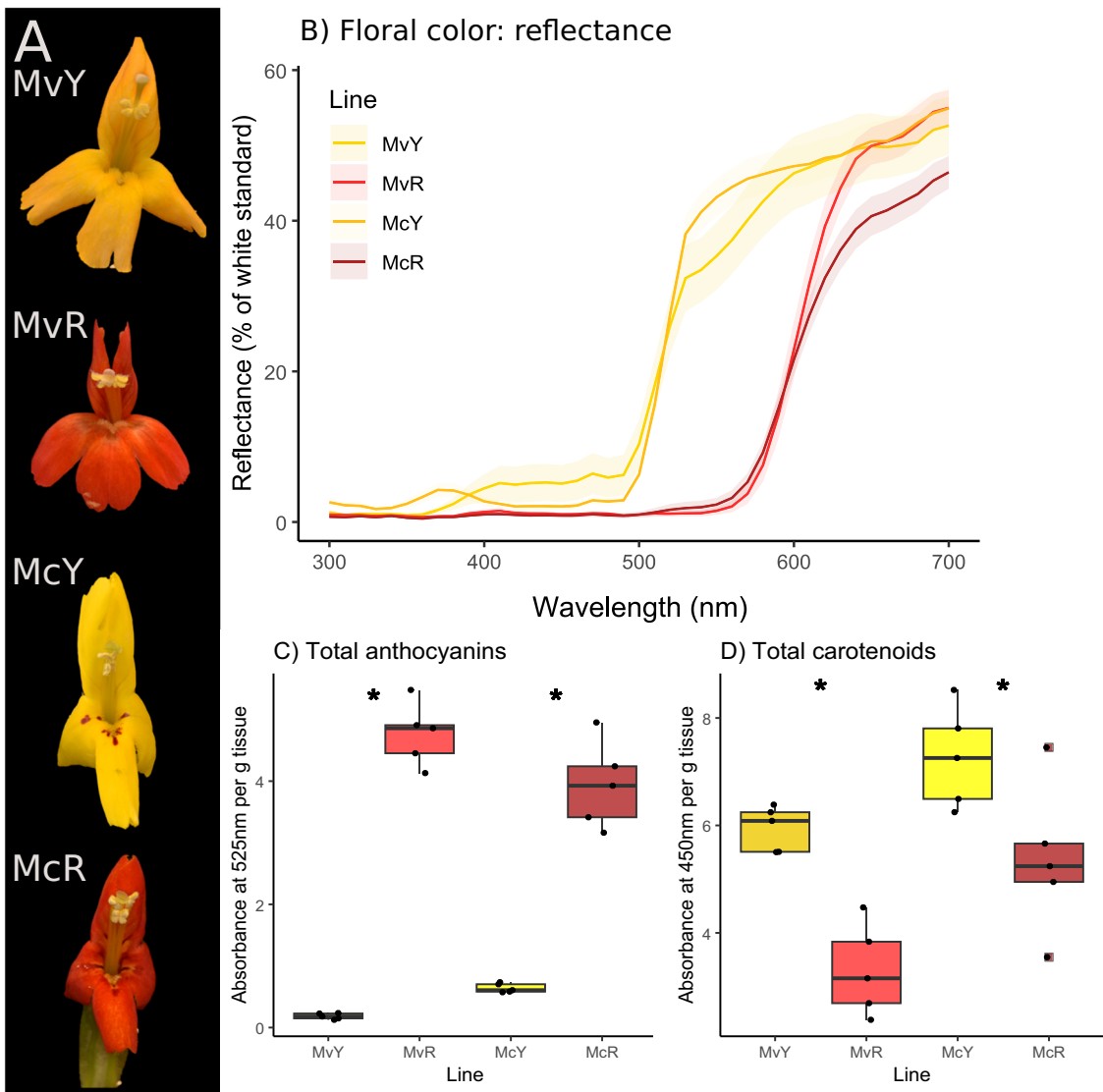

**Fig. 1 | Convergence in floral color shifts in two monkeyflower species: yellow flowers show increased carotenoids with low anthocyanins. A** Focal floral morphs (top to bottom): *M. verbenaceus*, MvY, yellow form, and MvR, red form; *M. cardinalis*, McY, yellow form, and McR, red form. **B** Spectral reflectance (color) of central petal lobes. Central line and shading shows mean reflectance +/− standard deviation; $N = 94$ flowers. **C** Total anthocyanins extracted from corolla tissues of each floral morph, measured by absorbance at 525 nm; $N = 20$ flowers (5 per line). **D** Total carotenoids, measured as absorbance at 450 nm, relative to average mass (g) of corolla per line; $N = 20$ flowers (5 per line). Boxplots show median (center bars), first and third quartiles (upper and lower hinges), points within 1.5*interquartile range of hinges (whiskers), outlying points (squares) and individual data points (dots). Asterisks denote significant differences between color morphs within species, based on 95% confidence intervals of least squares means with Tukey adjustments for multiple comparisons: Anthocyanins: MvR-MvY: $p < 0.0001$; McR-McY: $p < 0.0001$; Carotenoids: MvR-MvY: $p = 0.003$; McR-McY: $p = 0.03$. Fill colors denote plant line as labelled on axis. Details of statistics are located in the main text and Supplementary Table S2; sample sizes in Supplementary Table S1. Source data are provided as a Source Data file.

levels in McR than all other lines, including its yellow counterpart (Welch's $t$ test, $t = 4.208$ df = 4, $p = 0.0136$). Both isomers of antheraxanthin had a significant increase in MvY (Welch's $t$ test, $t = 7.240$ df = 7, $p = 0.0002$ for antheraxanthin 1, $t = 9.391$ df = 7, $p = 0.0001$ for antheraxanthin 2). In the last three steps of the pathway, a consistent trend is observed in which both yellow-flowered morphs accumulate significantly higher levels of violaxanthin (Welch's $t$ test, $t = 10.87$ df = 4, $p = 0.0004$ for MvY, $t = 6.299$ df = 6, $p = 0.0007$ for McY), neoxanthin (Welch's $t$ test, $t = 3.563$ df = 7, $p = 0.0092$ for MvY, $t = 4.108$ df = 7, $p = 0.0045$ for McY) and mimulaxanthin (Welch's $t$ test, $t = 8.517$ df = 7, $p = 0.0001$ for MvY, $t = 10.53$ df = 5, $p = 0.0001$ for McY) when compared to their red conspecifics, suggesting convergence in the final steps of biosynthesis and accumulation of carotenoids in both yellow morphs.

## Carotenoid biosynthesis genes

Following the evidence of convergence in carotenoid production, we next characterized the expression of genes involved in carotenoid biosynthesis. No significant differences were found in the initial steps of carotenoid biosynthesis in *M. verbenaceus*, whereas in *M. cardinalis* there is a significant difference in expression of *phytoene synthase-1* (*PSY1*), (Welch's $t$ test, $t = 8.161$ df = 3, $p = 0.0038$) and *phytoene synthase* (*PDS*)(Welch's $t$ test, $t = 6.032$ df = 3, $p = 0.0091$) between McR and McY. In the later steps of biosynthesis, both species show a significant increase in one homolog each of *β-carotene hydroxylase* (*BCH*) (Welch's $t$ test, $t = 4.520$ df = 3, $p = 0.0202$ for MvY, $t = 5.184$ df = 2, $p = 0.0353$ for McY) and *zeaxanthin epoxidase* (*ZEP*) (Welch's $t$ test, $t = 7.287$ df = 2, $p = 0.01$ for MvY, $t = 8.692$ df = 2, $p = 0.0130$ for McY)[49,50] which reflects the convergent increase in accumulation of

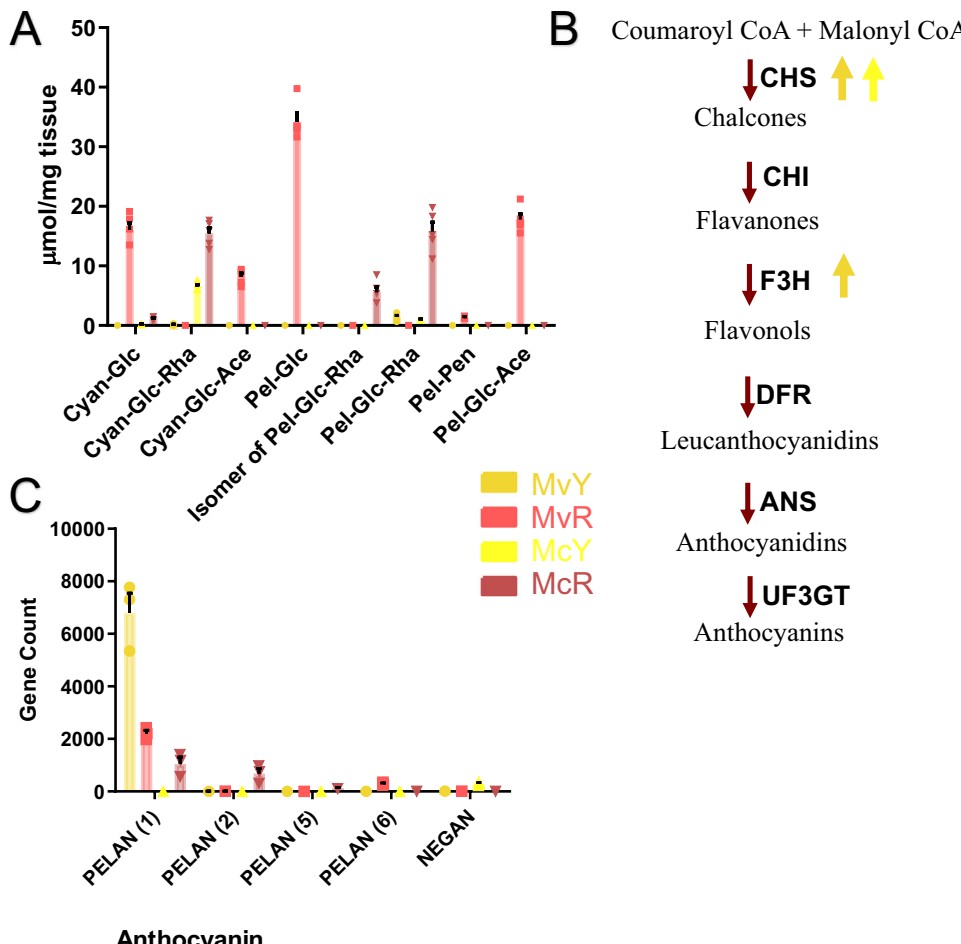

**Fig. 2 | Anthocyanin production is reduced in the yellow morphs. A** Individual anthocyanin levels in corollas of the four focal morphs as detected by UHPLC/MS ($n = 5$). Cyan Cyanidin, Pel Pelargonidin, Glc Glucoside, Rha Rhamnoside, Pen Pentose, Ace Acetate. **B** Core anthocyanin biosynthesis pathway. Genes responsible for each metabolic step are shown in bold next to each red arrow. Genes significantly differentially expressed ($p < 0.05$) in each yellow-flowered morph when compared to their red counterparts are denoted by an arrow corresponding with their color ID in the key in (**C**) Expression of anthocyanin biosynthesis regulators NEGAN and PELAN as filtered gene counts ($n = 3$). Two-way ANOVA used to confirm significance for arrows (Supplementary Table S6). Data are presented as mean values +/− SEM. Source data are provided as a Source Data file.

violaxanthin, neoxanthin and mimulaxanthin in both yellow morphs (Fig. 3B, C, Supplementary Table S10). Overall, these results show that the two yellow morphs are convergently increasing the production of the same carotenoid pigments by upregulating the same biosynthesis genes across both species.

## Transcriptome and whole genome variation

In order to better characterize whether both novel yellow morphs experienced similar genomic patterns of mutation accumulation, we characterized the number of within-species variants on each chromosome. All statistical analyses for the variant counts by chromosome window show the trends in variant accumulation are significantly different between species (Fig. 4, Supplementary Data 1). This suggests a lack of overall convergence in variant accumulation by chromosome region in the two red-yellow transitions.

Following from the work of[40], we extracted ITS sequences for *M. cardinalis* and *M. verbenaceus* from NCBI in order to visualize the phylogenetic relationship between these morphs using *Mimulus guttatus* as an outgroup[51]. The resulting tree demonstrates that the yellow morphs are indeed sister to their red counterparts for each of the three ITS sequences (Supplementary Fig. S1B). Moreover, it demonstrates how recent the emergence of these morphs is compared to the section *Erythranthe-Simiolus* (*M. guttatus*) split. These

analyses confirm that yellow morphs have arisen independently and very recently from within separate ancestrally red-flowered species. When viewing whole transcriptome expression patterns by NMDS (Supplementary Fig. S13), the morphs show clear species separation through MDS1 with distinction of colour morphs apparent through MDS2. Coupled with different patterns of genomic variation (Fig. 4), the color shifts are unlikely to be the product of introgression from one yellow morph into the other, in contrast to historical color locus introgression between multiple red species and morphs in *Mimulus* section *Diplacus*[29,30].

## Additional floral traits

To understand if recent shifts in floral color were accompanied by changes in additional floral traits (which could represent steps of smaller effect in an adaptive walk toward adaptation to a novel pollinator), we characterized floral scent, morphology, and nectar rewards for all four lines.

## Floral scent

*Mimulus verbenaceus* emitted considerable amounts of floral scent, while *M. cardinalis* produced very little scent (Fig. 5A). Scent composition was strongly different between morphs within species (Fig. 5B, C). We identified 39 compounds in both morphs of *M.*

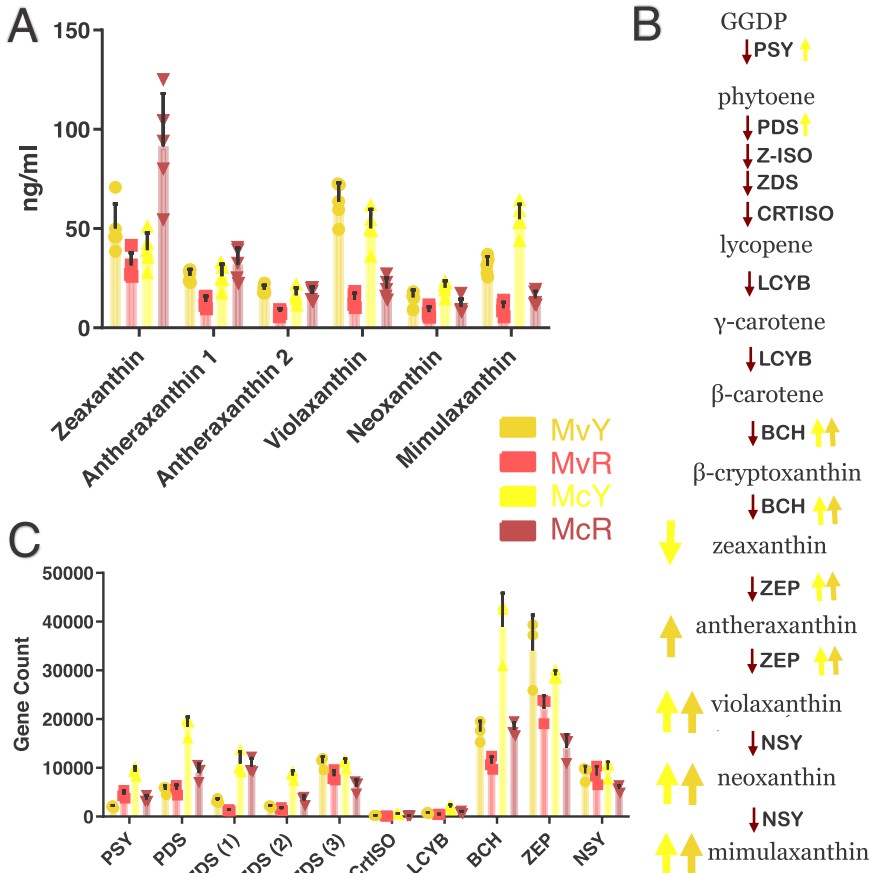

**Fig. 3 | Increased carotenoid accumulation shows evidence of convergence in yellow morphs. A** Individual carotenoid levels in whole corollas of the four focal morphs as detected by HPLC/MS ($n = 5$). **B** Carotenoid biosynthesis pathway. Genes responsible for each metabolic step are shown in bold next to each red arrow. Genes significantly differentially expressed ($p < 0.05$) in each yellow-flowered morph when compared to their red counterparts are denoted by an arrow corresponding with their color ID in the key in (**C**) Expression of carotenoid biosynthesis genes as filtered gene counts ($n = 3$). Two-way ANOVA used to confirm significance for arrows (Supplementary Table S10). Data are presented as mean values +/− SEM. Source data are provided as a Source Data file.

*verbenaceus* (Table 2), compared to ten compounds across both morphs of *M. cardinalis* (Table 3). *Mimulus verbenaceus* produced a high diversity of terpenoids (26 of 39 total compounds) and volatile composition was overall similar across color morphs (Fig. 5E), though terpenoid alcohols were mostly absent from red *M. verbenaceus*. Additionally, MvY had significantly higher total emission rates (2.32-fold) compared to MvR (Welch's *t* test, $t = 8.064$, df = 22.214, $p < 0.0001$). In contrast, McY and McR produced markedly different bouquets of floral VOCs (Fig. 5D), which they emitted overall at different, relatively low levels, with yellow morph total emissions approximately 1.68-fold higher than red morph emissions (Welch's *t* test, $t = 3.738$, df = 20.904, $p = 0.002$). None of the compounds (including unknowns) were found in common between the two species.

### Floral scent biosynthesis genes
Monoterpenoid synthase genes responsible for the biosynthesis of (*E*)-β-ocimene, limonene, and β-myrcene have been identified in section *Erythranthe*[36,52]. One already characterized *OCIMENE SYNTHASE (OS)* and five putative bifunctional *LIMONENE-MYRCENE SYNTHASEs (LMS)* were identified in the genome of MvR based on homology to published *M. lewisii* sequences[36]. *OS* was expressed only in *M. cardinalis*, where it is known to be non-functional in McR (=CE10) (Supplementary Fig. S14), and no morph produces (*E*)-β-ocimene. The five different *LMS* candidates were expressed to different degrees (Supplementary Fig. S14), despite limonene's absence from all four. Although both

morphs of *M. verbenaceus* produce β-myrcene, the lack of concurrent limonene production indicates that either a different gene is responsible for β-myrcene production or an *LMS* mutation has removed limonene production in *M. verbenaceus*. Neither product of *LMS* is produced in *M. cardinalis*, although all five *LMS* candidates are expressed. These putative *LMS* genes may be responsible for the production of other terpenes, or may have become pseudogenes, as previously found in McR (=CE10) *LMS*[36].

### Floral morphology and nectar rewards
We measured floral morphology traits involved in pollinator attraction (floral display height and width), mechanical fit (corolla tube length, corolla opening height and width, and herkogamy) and nectar reward quality (volume and percent sugar). Morphological traits varied significantly among species (MANOVA: approximate $F_{1,97} = 81.628$, $p < 0.0001$) and between lines nested within species (MANOVA: approximate $F_{2,97} = 18.613$, $p < 0.0001$). NMDS ordination of morphological traits revealed overall separation between the two species, but little separation between MvR and MvY, in contrast to clear separation between McR and McY (Fig. 6A).

All individual morphological traits varied significantly among species ($p \leq 0.001$; Supplementary Table S3; Supplementary Fig. S4), with the exception of herkogamy ($p = 0.1$). Corolla length varied among morphs nested within species ($\chi^2_{2, 117} = 53.99$, $p < 0.0001$), with MvY having significantly longer corollas than MvR, while McY had significantly shorter corolla tubes than McR (Fig. 6B). Opening height

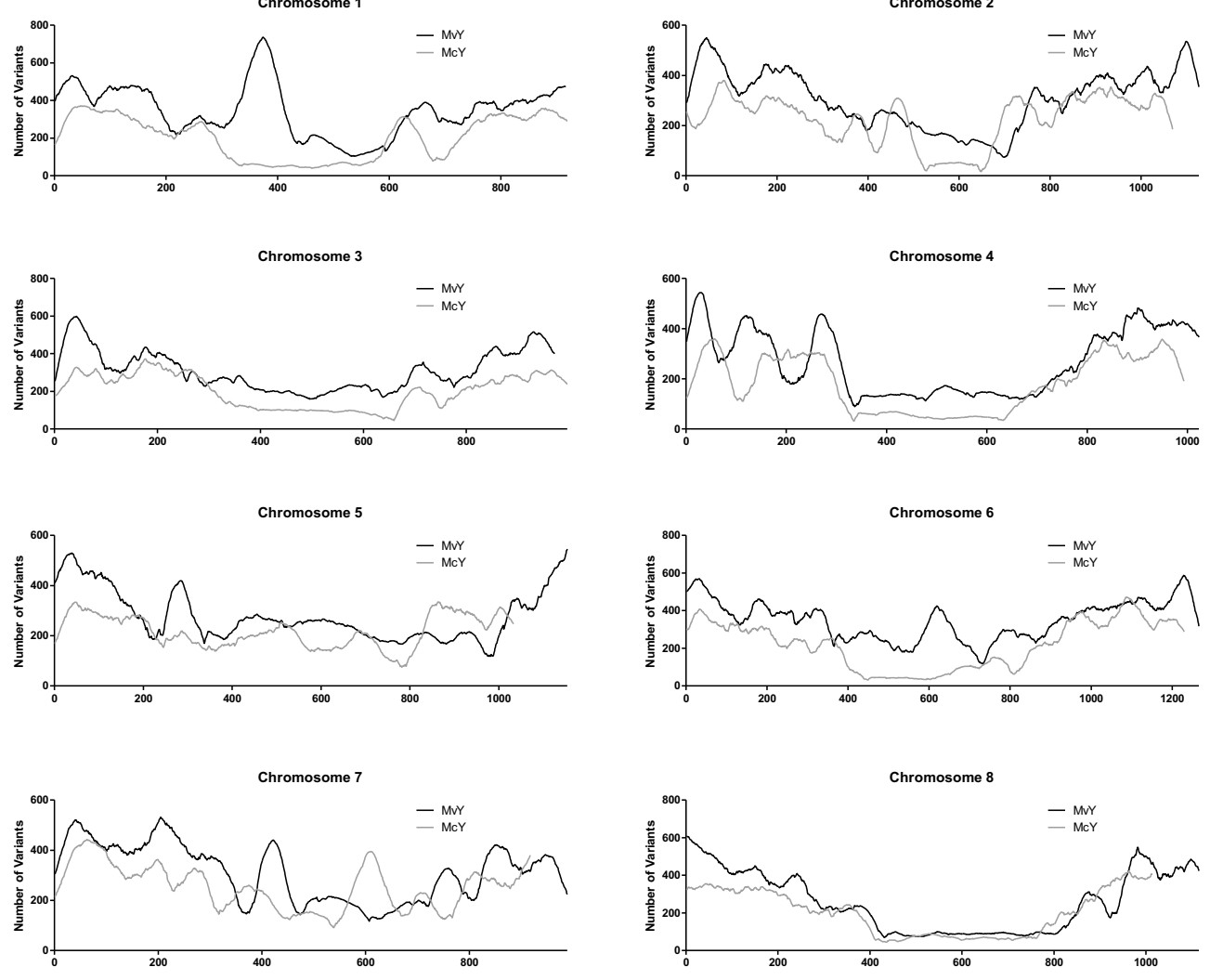

**Fig. 4 | Trends in variant accumulation across genomes are divergent between yellow morphs.** Whole genomes of yellow-flowered morphs MvY and McY were mapped to the published genomes of their respective red counterparts and the number of variants called using bcftools. Chromosomes were divided into 30 kb windows with the number of variants detected per window plotted for each chromosome and the two species overlaid. Source data are provided as a Source Data file.

and width both varied significantly between morphs within species (height: $\chi^2_{2, 98} = 65.635$, $p < 0.0001$; width: $\chi^2_{2, 98} = 57.03$, $p < 0.0001$). For both species, yellow-flowered lines had significantly larger openings than their conspecific red-flowered lines in both opening height (Fig. 6C) and width (Supplementary Fig. S4). Display height varied significantly among lines within species ($\chi^2_{2, 99} = 58.16$, $p < 0.0001$; Supplementary Fig. S4), with MvY significantly greater then MvR, the only significant pairwise species comparison. Display width varied significantly among lines within species ($\chi^2_{2, 99} = 11.29$, $p = 0.004$; Supplementary Fig. S4); however, no pairwise comparisons were significantly different. Herkogamy (within-flower stigma-anther separation) did not vary between species but did vary significantly among lines nested within species ($\chi^2_{2, 63} = 116.34$, $p < 0.0001$), with both yellow flowered lines displaying greater herkogamy than their red-flowered conspecifics (Fig. 6D).

Nectar volume per flower varied between species ($\chi^2_{1, 93} = 48.94$, $p < 0.0001$) but not between lines within species ($\chi^2_{2, 93} = 2.28$, $p = 0.32$; Supplementary Fig. S4). Percent sugar varied between species ($\chi^2_{1, 93} = 74.25$, $p < 0.0001$) and among lines ($\chi^2_{2, 93} = 17.38$, $p = 0.0002$), with the only significant pairwise comparison being in *M. verbenaceus*, where MvY had a higher sugar concentration than MvR (Supplementary Fig. S4).

### Floral trait category correlation analysis

Floral traits may be correlated within trait categories, suggesting category-specific floral integration. Distributions of the absolute value of intra-category Pearson correlation coefficients (PCCs) were higher in *M. verbenaceus* than *M. cardinalis* for volatile ($F_3 = 7.330$, $p = 7 \times 10^{-5}$) and nectar trait categories (Supplementary Fig. S5). No differences were seen between species for color ($F_3 = 1.727$, $p = 0.16$) or morphology ($F_3 = 0.866$, $p = 0.46$) trait categories, indicating that these are equally integrated between the two species. As terpenoid volatiles share the same biosynthetic precursors as carotenoid pigments[53], we also calculated correlations between total terpenoid emission and total carotenoids, with species grouped by color morph. These were significantly negatively correlated in the two grouped red lines ($R = -0.866$, $p = 0.0012$), suggesting a biochemical tradeoff, but were not significantly correlated in yellow lines ($R = 0.111$, $p = 0.76$), despite the higher production of both in yellow morphs.

Individual floral trait correlation analysis - Floral traits may also be individually correlated, even across trait categories. Traits were generally more strongly correlated in *M. verbenaceus* (Supplementary Figs. S6 and S7) than in *M. cardinalis* (Supplementary Figs. S8 and S9), which demonstrated very little inter-trait correlation, even within trait categories. There were 34 traits in McY yielding 544 pairwise

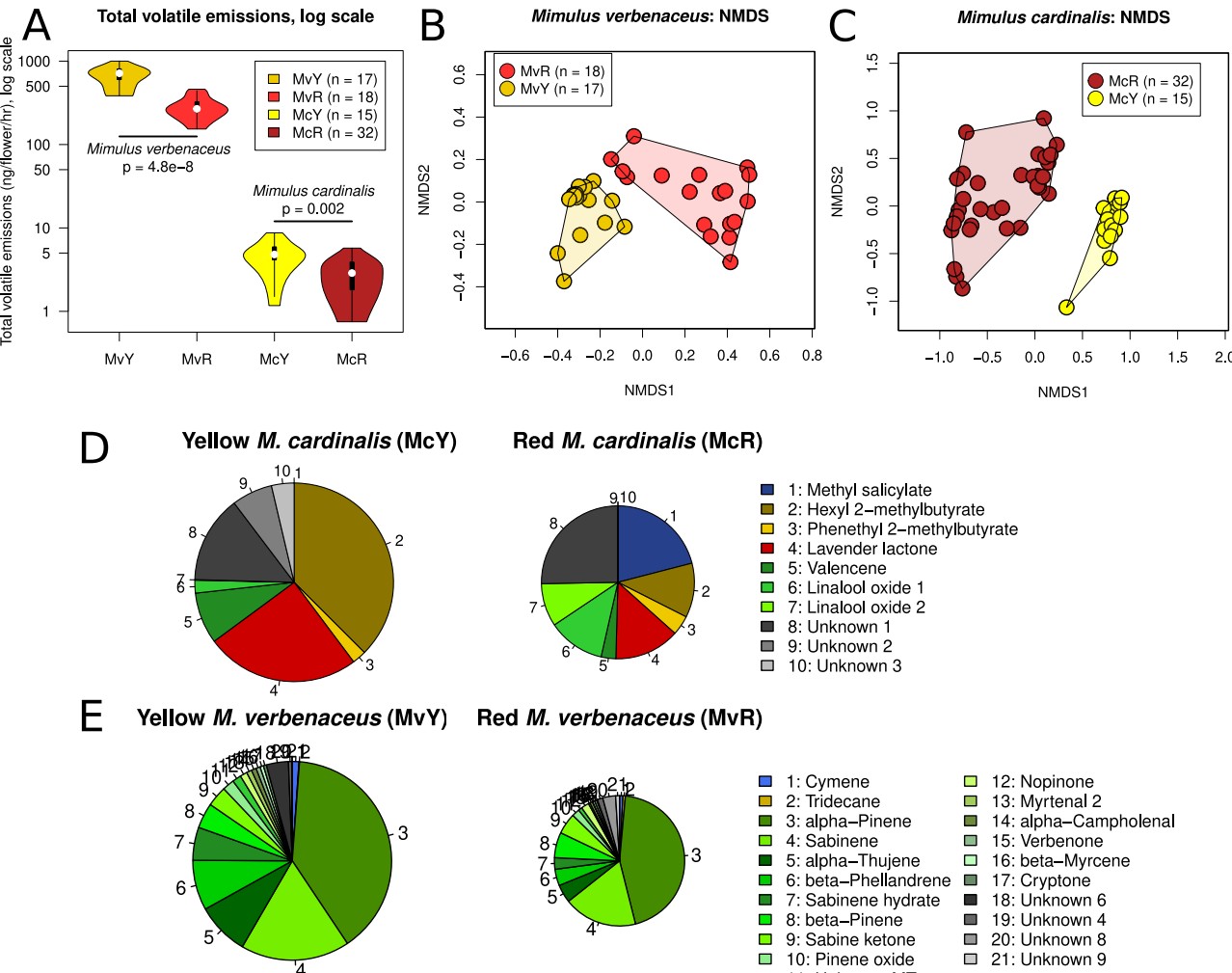

**Fig. 5 | Floral scent emissions vary in total volatiles (both species) and global composition of VOCs (*M. cardinalis*) between red and yellow conspecific floral morphs. A** Comparison of total volatile emissions of the four species-color morph lines, with the y-axis on a log scale. Welch Two Sample two-sided t-test for each species (*M. verbenaceus*: $t = 8.064$, df = 22.214, $p = 4.8e{-}8$; *M. cardinalis*: $t = 3.738$, df = 20.904, $p = 0.002$). Internal violin plot bars extend to the closest data point 1.5 times the interquartile box length; black box length represents the interquartile range; white dot indicates the median; upper and lower borders represent the maximum and minimum data points. $N = 17$ (MvY), 18 (MvR), 15 (McY), 32 (McR). **B** NMDS plot of floral scent samples in *M. verbenaceus*. **C** NMDS plot of floral scent samples in *M. cardinalis*. **D** Floral scent composition in color morphs of *M. cardinalis*. Only compounds contributing at least 1% of total emission in either morph are shown and area of the pie is relative to total emission of the yellow morph. **E** Floral scent composition in color morphs of *M. verbenaceus*, with compound choice and area as in (**D**). MT monoterpenoid. Note that no unified NMDS plot of *M. verbenaceus* and *M. cardinalis* is possible due to a lack of volatile sharing between species. Source data are provided as a Source Data file.

comparisons, of which 34 (6.25%) were significantly correlated. McR had 32 traits and 496 comparisons, of which 19 (3.83%) were significant. MvY had 63 traits and 1953 comparisons, of which 288 (14.75%) were significant. MvR had 64 traits and 2016 comparisons, of which 317 (15.72%) were significant.

**Pollinator perception of color and scent**

We visualized how bees would perceive the four lines' corolla colors by plotting central petal lobe reflectance in a trichromatic model of honeybee visual space (Fig. 7A[54]), which revealed that both red-flowered lines overlapped with each other and showed low contrast from a green background. MvY and McY appear distinct from both from each other and their red conspecifics, and also fall further outward on the axes, indicating greater contrast with a green background compared to the red-flowered morphs.

To test scent perception, we performed electroantennography (EAG) to assess the physiological responses of *Bombus terrestris* ssp. *audax* (hereafter *B. terrestris*), a novel bumblebee pollinator with broad floral preferences[55] and similar size to native bumblebees observed on these species[33,45,56], to floral scent of all morphs and several controls. Bumblebees' electroantennographic responses varied significantly among stimuli ($\chi^2 = 78.28$, df = 6, $p < 0.0001$), and all four floral scent extracts elicited a significantly greater response than the air-only negative control (Fig. 7B, Supplementary Table S4), suggesting these floral scents can be perceived by bumblebee pollinators. However, antennal responses to floral scent extracts were only significantly greater than to extraction solvent alone for the two *M. cardinalis* morphs (McR $t = 3.32$ df = 48.4 $p = 0.03$; McY $t = 5.13$ df = 48.4 $p = 0.0001$), were marginally greater for MvY ($t = 3.03$ df = 48.5 $p = 0.06$), and were not significantly different for MvR ($t = 1.68$ df = 48.4 $p = 0.63$). Additionally, only one pairwise comparison among the different floral scents was significantly different (McY response was significantly higher than MvR, $t = -3.45$ df = 48.4 $p = 0.02$). Thus, while bumblebees could perceive the floral scents of all floral morphs, we did not find evidence that morphs were likely to vary in conspicuousness to bees based on their floral scent profiles

**Table 2 | Volatile emissions (ng/flower/hour ± standard error of the mean) of each volatile identified from *Mimulus verbenaceus***

| Compound name | Type | Kovats Retention Index | Yellow *M. verbenaceus* MvY (*n* = 17) | Red *M. verbenaceus* MvR (*n* = 18) |
|---|---|---|---|---|
| Total emissions (summed across all compounds) | – | – | **675.830 ± 43.538** | **291.169 ± 19.487** |
| Cymene[A] | Aromatic | 1021 | **7.096 ± 0.860 (16)** | **2.374 ± 0.361 (17)** |
| α-Thujene[A] | Terpenoid | 923 | **55.323 ± 5.055 (17)** | **12.770 ± 2.420 (18)** |
| α-Pinene[A] | Terpenoid | 929 | **254.941 ± 18.331 (17)** | **126.776 ± 8.250 (18)** |
| Sabinene[A] | Terpenoid | 971 | **114.602 ± 9.888 (17)** | **51.895 ± 4.510 (18)** |
| β-Pinene[A] | Terpenoid | 972 | **26.593 ± 3.216 (17)** | **17.861 ± 0.767 (18)** |
| β-Phellandrene[A] | Terpenoid | 1025 | **52.585 ± 3.663 (17)** | **11.589 ± 1.592 (18)** |
| Sabinene hydrate[A] | Terpenoid | 1063 | **34.796 ± 5.026 (15)** | **8.154 ± 1.800 (11)** |
| Pinene oxide[A] | Terpenoid | 1092 | **12.466 ± 1.461 (17)** | **5.428 ± 0.545 (17)** |
| Unknown MT[C] | Terpenoid | 1095 | **9.281 ± 1.130 (17)** | **2.636 ± 0.337 (17)** |
| Nopinone[A] | Terpenoid | 1133 | 7.468 ± 0.526 (17) | 5.563 ± 1.046 (18) |
| Sabine ketone[B] | Terpenoid | 1153 | 21.484 ± 2.918 (17) | 14.803 ± 2.303 (18) |
| *M. verbenaceus* Unk 3[C] | Unknown | 1273 | 3.348 ± 0.448 (16) | 3.922 ± 0.983 (15) |
| *M. verbenaceus* Unk 5[C] | Unknown | 1392 | **23.187 ± 3.656 (16)** | **0.521 ± 0.357 (2)** |
| *M. verbenaceus* Unk 7[C] | Unknown | 1942 | **0.537 ± 0.207 (8)** | **9.071 ± 1.868 (17)** |

Only volatiles above 1% of total emissions are shown; see Supplementary Table S13 for full list. Numbers in parentheses after each emission value indicate the number of samples a compound was found in from that line (total sample numbers for each line are in the table header). Under the Compound Name header, superscript letters: A: compound identity validated using authentic reference standards; B: compound identity validated using published Kovats Retention Indices, our calculated Kovats Retention Indices, and NIST Library spectrum matching; C: compound identity could not be validated and compound is listed as an unknown. Values in bold text differed significantly between red and yellow morphs. Note for Cymene: m-/o-/p- structure could not be differentiated with standards.
*MT* monoterpenoid, *Unk* unknown.

**Table 3 | Volatile emissions (ng/flower/hour ± standard error of the mean) of each volatile identified from *Mimulus cardinalis***

| Compound name | Type | Kovats Retention Index | Yellow *M. cardinalis* McY (*n* = 15) | Red *M. cardinalis* McR (*n* = 32) |
|---|---|---|---|---|
| Total emissions (summed across all compounds) | – | – | **5.036 ± 0.493** | **2.985 ± 0.240** |
| Methyl salicylate[A] | Aromatic | 1191 | **Absent (0)** | **0.626 ± 0.047 (29)** |
| Hexyl 2-methylbutyrate[A] | FAD | 1238 | **1.888 ± 0.132 (15)** | **0.343 ± 0.032 (29)** |
| Phenethyl 2-methylbutyrate[A] | FAD | 1485 | 0.115 ± 0.028 (10) | 0.124 ± 0.015 (25) |
| Lavender lactone[B] | Lactone | 1041 | **1.265 ± 0.160 (15)** | **0.412 ± 0.069 (22)** |
| Linalool oxide isomer 1[A] | Terpenoid | 1070 | **0.105 ± 0.023 (10)** | **0.360 ± 0.036 (30)** |
| Linalool oxide isomer 2[A] | Terpenoid | 1086 | **0.005 (1)** | **0.272 ± 0.045 (27)** |
| Valencene[A] | Terpenoid | 1482 | **0.421 ± 0.093 (13)** | **0.093 ± 0.043 (7)** |
| *M. cardinalis* Unk 1[C] | Unknown | 950 | 0.717 ± 0.093 (13) | 0.755 ± 0.181 (13) |
| *M. cardinalis* Unk 2[C] | Unknown | 1052 | **0.334 ± 0.074 (10)** | **Absent (0)** |
| *M. cardinalis* Unk 3[C] | Unknown | 1687 | **0.185 ± 0.018 (14)** | **Absent (0)** |

All volatiles are shown, as all comprise more than 1% of total emission for either morph. Numbers in parentheses after each emission value indicate the number of samples a compound was found in from that line (total sample numbers for each line are in the table header). Under the Compound Name header, superscript letters: A: compound identity validated using authentic reference standards; B: compound identity validated using published Kovats Retention Indices, our calculated Kovats Retention Indices, and NIST Library spectrum matching; C: compound identity could not be validated and compound is listed as an unknown. Values in bold text differed significantly between red and yellow morphs.
*FAD* Fatty-acid derived compound, *Unk* unknown.

alone. This is despite the overall much higher total volatile emission in *M. verbenaceus*, suggesting that total volatile emission may only need to reach a certain level for antennal perception. To explore the response of other potential novel pollinators to floral scent, we performed equivalent EAG experiments with *Manduca sexta* hawkmoths. These revealed similar patterns to those of bumblebees, with moths displaying significantly higher antennal responses to scents of all floral morphs compared to controls, but no significant variation in response among floral morphs (Supplementary Fig. S10, Supplementary Table S5).

## Pollinator preference: bumblebee choice
Finally, we tested behavioral responses of novel bumblebee pollinators to yellow and red morphs in controlled within-species comparisons.

When ten naïve *Bombus terrestris* workers were offered equal numbers of red and yellow *M. verbenaceus* flowers *M. verbenaceus* in a single-bee experimental array, they made a total of 89 floral visits (Supplementary Fig. S6). Fifty-nine (66.3%) of these visits were to MvY and 30 (33.7%) were to MvR, revealing a significant preference for the yellow-flowered morph ($\chi^2_1 = 9.45$, $p = 0.002$). For *M. cardinalis*, bees made a total of 150 visits over ten trials, visiting McY significantly more frequently, with 99 visits (66%) to McY and 51 visits (34%) to McR ($\chi^2_1 = 15.36$, $p < 0.0001$). Despite this clear preference for yellow in total number of choices across both species, bees were not more likely to choose a yellow flower for their first (naïve) choice (in both species, six first choices were to yellow and four to red; $\chi^2_1 = 0.4$, $p = 0.53$).

Bumblebees preferred yellow flowers of *M. verbenaceus* over red in metrics of total visit number ($\chi^2_1 = 9.44$, $p = 0.002$; Fig. 7C) and

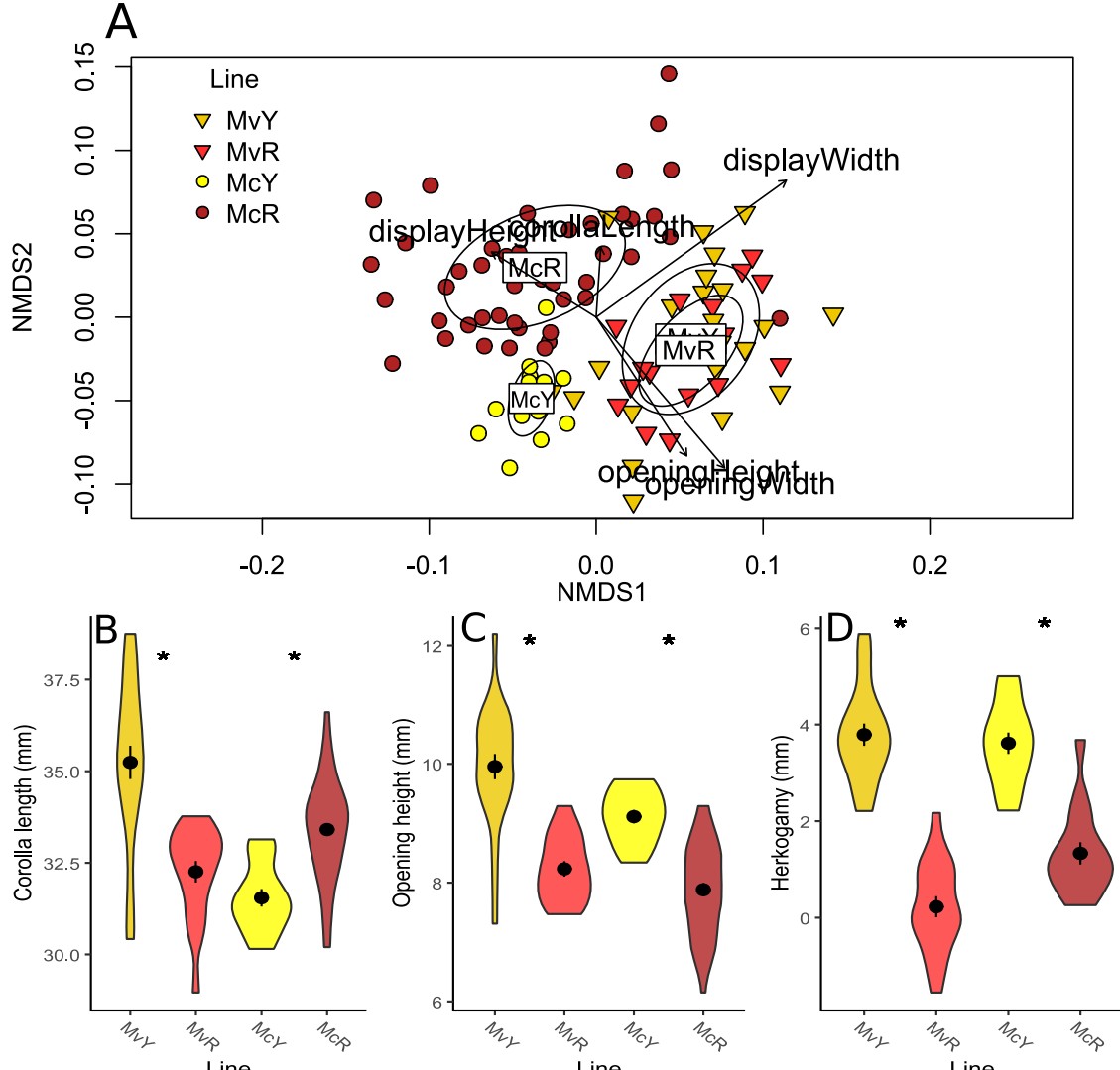

**Fig. 6 | Floral morphology varies among color morphs, potentially impacting pollinator fit. A** NMDS ordination of floral morphological traits by line. Line centroid is labeled, with circles showing standard deviation (*N* = 101 flowers). Individual trait loadings with *p* < 0.05 based on 1000 permutations (envfit()) are plotted on the ordination for visualization (corolla length: *p* = 0.02; all other traits: *p* < 0.001) (**B**–**D**) Violin plots of selected morphological traits by line, with mean (central point) +/− standard error (bars); asterisks denote significant pairwise differences between morphs within species based on 95% confidence intervals of least squares means with Tukey adjustments for multiple comparisons (*p* < 0.05): **B** Corolla length (mm): *N* = 120 flowers; MvR-MvY: *p* < 0.0001; McR-McY: *p* = 0.0012; **C** Opening height of corolla tube (mm): *N* = 101 flowers; MvR-MvY: *p* < 0.0001; McR-McY: *p* = 0.0002; **D** Herkogamy (distance between anthers and stigma; mm): *N* = 66 flowers; MvR-MvY: *p* < 0.0001; McR-McY: *p* = 0.0002. Fill colors denote plant line as labelled on axis. Details of statistics are located in the main text and Supplementary Table S3; sample sizes in Supplementary Table S1. Source data are provided as a Source Data file.

total handling time ($\chi^2_1$ = 11.74, *p* = 0.0006), but only marginal evidence for number of flowers probed ($\chi^2_1$ = 3.65, *p* = 0.056; Fig. 7D). For *M. cardinalis*, bees significantly preferred yellow flowers in metrics of total visit number ($\chi^2_1$ = 15.125, *p* = 0.0001), number of flowers probed ($\chi^2_1$ = 10.07, *p* = 0.002), and total handling time ($\chi^2_1$ = 74.07, *p* < 0.0001). Worker bees did not always closely contact the anthers and stigma of flowers while nectaring (Fig. 7E) and were observed to have difficulty handling flowers and accessing floral rewards, particularly in the longer-tubed MvY, suggesting incomplete morphological adaptation to bumblebee pollination. Bees commonly crawled around the back of corollas in an attempt to nectar rob or check for existing robbing holes. When bees made persistent efforts to enter corollas to reach nectar at the base of floral tubes this often resulted in considerable damage to flowers, including bruising and ripping of corollas (Fig. 7E).

## Discussion

We characterized two yellow-flowered morphs, established as stable populations in two otherwise red-flowered hummingbird-pollinated species, *Mimulus verbenaceus* and *M. cardinalis*, across a suite of floral traits important to pollinator attraction and fit and examined genetic differences underlying these repeated color transitions. We confirmed the previously hypothesized loss of anthocyanins in petal lobes[33] and involvement of the gene *PELAN* in these shifts[42]. Surprisingly, both yellow morphs also increase their production of the same carotenoid pigments using convergent upregulation of the same carotenoid biosynthesis genes via independent genetic changes. These recent, intraspecific transitions in color were accompanied by changes in additional traits related to pollinator attraction and fit. Both yellow forms increased overall floral volatile emissions and had wider, less restrictive corolla openings and greater herkogamy. Nectar rewards

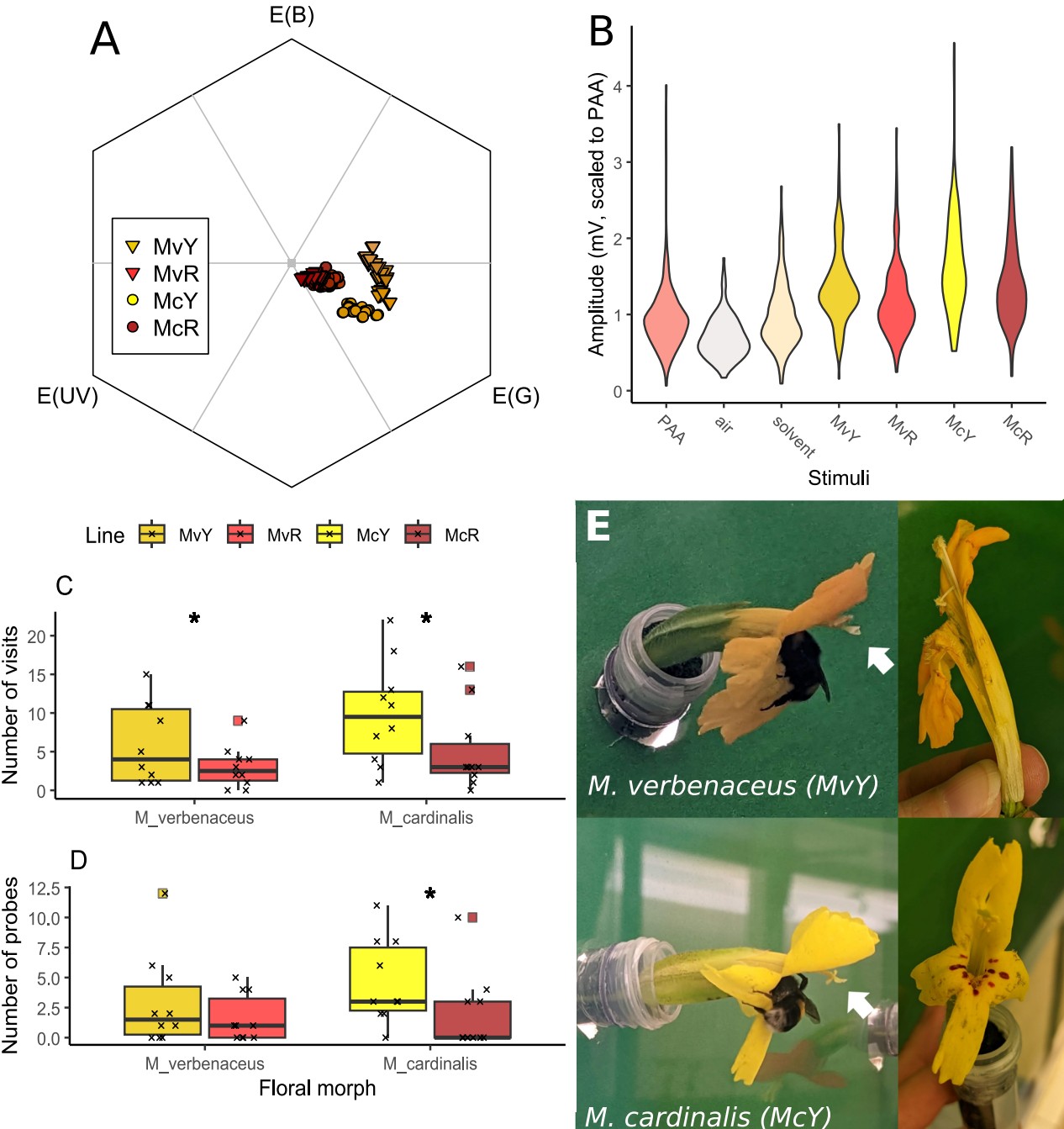

**Fig. 7 | Bumblebees prefer yellow flowers: increased attraction despite poor fit.**
**A** Floral color perception by bees. Central petal lobe reflectance measurements modelled in bee visual space, where the center point represents low visual contrast to a green vegetative background. $N = 283$ reflectance measurements taken from 94 flowers. **B** EAG responses to scent stimuli, including phenylacetaldehyde (PAA, a floral VOC known to be detectable by *Bombus terrestris*), air (negative control), extraction solvent alone, and floral scent extractions of MvY, MvR, McY, and McR. $N = 9$ bees, 1230 presentations total: PAA: 315 presentations; air: 153, solvent: 153, MvY: 150, MvR: 153, McY: 153, McR: 153. Fill colors denote stimulus as labelled on axis. **C, D** Bee preference metrics in pairwise choice experiments between conspecific pairs of color morphs ($N = 10$ trials per species comparison). Boxplots show median value (center

bars), first and third quartiles (upper and lower hinges), points within 1.5*interquartile range of hinges (whiskers), outlying points (squares), and individual data points (x's). Asterisks(*) denote significant differences ($p < 0.05$) between morphs within species based on TypeII Wald chi-square tests (see Results text for detailed statistics). **C** Number of visits (MvY-MvR: $p = 0.002$; McY-McR: $p = 0.0001$); **D** number of flowers probed: (MvY-MvR: $p = 0.056$; McY-McR: $p = 0.002$). **E** Bumblebees probing yellow morphs, often with poor contact to anthers and stigma (left, arrows), and resulting damage to flowers (right). Note corolla tube split open from bee entering corolla in *M. verbenaceus* (top-right photo). Details of statistics are located in the main text and Supplementary Table S4; sample sizes in Supplementary Table S1. Source data are provided as a Source Data file.

were largely similar across color morphs within species. Bees were more likely to visually perceive the yellow flowers of both species and could detect the floral scent of all floral morphs. Collectively, these floral trait differences resulted in a clear preference of naïve bumblebees for yellow morphs in both species, despite difficulty handling

flowers, which likely reflects the incipient nature of this intraspecific pollinator shift. Overall, this study reveals that repeated transitions in floral color within two closely related hummingbird-pollinated species may follow largely convergent adaptive walks across a suite of floral traits in an incipient shift to bumblebee pollination, which strongly

impacts the potential for reproductive isolation and adaptive evolution.

## Convergence in floral pigments and their genetic basis

We found that shades of the two yellow morphs result not only from a near lack of anthocyanins as expected[42,45], but also from a simultaneous increase in carotenoid pigments. The yellow morphs in both species have independently increased production of the same three carotenoid pigments (violaxanthin, neoxanthin, and mimulaxanthin) via upregulation of the same two carotenoid biosynthesis genes (*BCH* and *ZEP*), suggesting convergence in these independent color shifts. Variation in anthocyanin accumulation was reflected in the expression patterns of known anthocyanin regulators. *PELAN* is known to be responsible for anthocyanin accumulation in petal lobes of McR (CE10) and is deleted from the genome of McY (SM)[42]. The high expression of this locus in MvY is surprising, given the near absence of anthocyanins in this line. However, studies in *Lotus*[57], *Fagopyrum*[58] and most recently *Lysimachia*[28] have found that differential regulation of gene paralogs by the same MYB transcription factors mediated via promoter sequence polymorphisms can underlie floral color polymorphisms. We propose a similar case here, where *PELAN* expression in MvY may play a suppressive role in anthocyanin accumulation, though further study is required.

Floral color transitions in multiple related species that involve the same genes have previously been attributed to adaptive introgression of alleles mediated by gene flow[30] or to ancestral polymorphism predating speciation[28]. If either of these processes were responsible here, we would expect to find shared patterns of mutations in involved loci, which was not the case. In contrast, we find evidence that differential expression of the same loci results from independent mutations, consistent with distinct patterns of sequence variation among color morphs throughout their genomes (Fig. 4). Analysis of additional variation found between focal morphs at genomic and transcriptomic levels is outside the scope of this study but opens opportunities for further exploration in this system.

## Bird-to-bee pollinator shifts

Red floral pigmentation is often associated with hummingbird pollination[59,60], though recent work suggests red flowers may function less to attract hummingbirds[61] than to deter bumblebees[62]. We also note that the UV bullseye pattern seen in yellow morphs may make them more visible to bees, whose eyes have receptors sensitive to UV wavelengths[63]. Shifts from blue-purple bee-pollinated flowers to red hummingbird-pollinated flowers are considered strongly directional[64,65], and reversals from hummingbird to bee pollination are uncommon[66]. However, given multiple examples of transitions between red and yellow flowers[30,67–69] and reported associations of bumblebees with yellow over red flowers[13,45,70], perhaps shifts from red to yellow flowers as here may represent an avenue to re-gain ancestral attraction to bumblebees or other insects[64,67].

The potential for these repeated transitions to yellow flowers in *M. verbenaceus* and *M. cardinalis* to represent shifts to bumblebee pollination is consistent with changes in other floral traits. Both yellow morphs had increased scent emission, which could reflect selection from pollinators such as bees that use scent to forage, unlike hummingbirds[71,72], although this might require potentially unlikely (but possible[73]) gain-of-function mutations. Both yellow floral forms were characterized by wider, less restrictive tube openings, which might function to allow easier access to nectar by bees. This increased access may be further aided in McY by shorter corolla tubes compared to McR (though the opposite was observed in *M. verbenaceus*, discussed below), as well as the presence of bee-orienting nectar guides[74,75]. This combination of traits particular to McY could contribute to the significant increase in probed yellow flowers in this species. Nectar volume did not vary among color morphs in either species (though in *M. verbenaceus*, yellow flowers had higher sugar content than red), which is noteworthy as rewards are important in determining pollinator preference[76,77].

In addition to color, floral scent is a key trait involved in pollinator attraction[78–80]. Based on our EAG analysis, workers of *Bombus terrestris* are capable of detecting the floral scents of all morphs, though we did not see a greater response from bee antennae to either yellow morph. It is possible that the presented floral volatile stimuli are equivalently ecologically relevant to bumblebees, resulting in no response difference despite significant volatile composition differences, as seen in other studies[81]. It is also possible that our EAG analysis was complicated by the inadvertently low concentration of the positive control stimulus. We note that our scent profile of McR (=CE10) differs from that described in Byers et al.[35], which is likely a result of our more stringent filtering of compounds and a more sensitive GCMS system.

Overall, the most compelling evidence for a potential shift to bee pollination in this system is the strong preference of the bees themselves for yellow morphs. We found significant preference by bumblebees for yellow in both species, with a notably consistent 2:1 ratio of all visits to yellow versus red flowers across species (Supplementary Fig. S11). Nonetheless, during the experiment, bees were observed to have difficulty handling flowers. In particular, bumblebees entered the corolla in multiple orientations, often poorly contacting floral organs (Fig. 7E), which would hinder pollen transfer efficiency. Bees were also observed attempting to nectar rob, and often struggled to fit inside corolla tubes to access nectar, occasionally resulting in severe corolla damage when bees persisted in trying to enter (Fig. 7E, particularly in the longer-tubed MvY).

Given that these floral morphs represent recent transitions below the species level, we suggest they represent incipient pollinator shifts which may occupy a fitness valley between peaks of floral phenotypes optimally adapted to either hummingbird or bee pollination. Based on the strong increase in bumblebee attraction observed in both yellow morphs, we hypothesize that convergent shifts in floral color (loss of anthocyanins and increased carotenoids) represent mutations of large effect at the beginning of an ongoing adaptive walk with strong effects on attraction of a novel bumblebee pollinator, while additional traits likely reflect smaller-effect steps contributing further to attraction (increased scent emission), fit (wider corolla openings), and positioning (nectar guides in McY) of bee pollinators. We speculate that additional changes in floral traits characteristic of bumblebee-pollinated flowers, such as landing platforms and less tubular corollas (characteristic of *M. lewisii*, the bee-pollinated member of section *Erythranthe* sister to *M. cardinalis*) would represent further steps needed to complete the adaptive walk to an optimal bee-pollinated phenotype. Future research into this putative pollinator shift should extend to natural populations by assessing allelic variation and potential signatures of selection of identified loci, along with characterizing variation in floral traits beyond the inbred reference lines used here.

With this in mind, we acknowledge that pollinators besides hummingbirds and bumblebees may pollinate and exert selection on these flowers in natural habitats. In particular, we hypothesize that the yellow morph of *M. verbenaceus* could reflect an incipient shift to hawkmoth pollination, given its pale coloration, long corolla tubes, increased floral scent emission, and lack of nectar guides[82]. While speculative, the potential for hawkmoth pollination is consistent with our EAG results for the hawkmoth *Manduca sexta*, as hawkmoths are capable of detecting scent profiles of all morphs (Supplementary Fig. S6), and *Manduca sexta* preferred yellow flowers over red in controlled experiments with *Mimulus*[83]. Shifts from hummingbird to moth pollination have occurred repeatedly in other systems (e.g. *Aquilegia*[64]), and may reflect that many hawkmoths, including the regionally common *Hyles lineata*, are generalists[13,84], and are often underreported in diurnal-only pollinator observations (e.g. ref. 45). Alternatively, observed floral trait variation could reflect selection

from other biotic or abiotic drivers[85] or drift, which is expected to be stronger at range edges[86] where both yellow morphs occur. While other explanations of floral trait variation warrant further study, our reported twofold preference among bumblebees for yellow morphs in both species provide convincing evidence that these floral traits are likely to be subject to pollinator-mediated selection. Future research into the native pollinators and other environmental conditions of yellow morphs of both species in situ, along with nearby red-flowered populations, is needed to more fully investigate the eco-evolutionary conditions which may favor a pollinator shift within these species.

In this study, we document that two intraspecific floral color transitions result from convergent patterns of carotenoid accumulation, in addition to loss of anthocyanins, both of which involve independent mutations in a shared set of pigment-related genes. These changes in floral color are accompanied by variation in a suite of additional floral traits, with implications for the attraction, mechanical fit, and reward access of a novel pollinator. The novel yellow morphs in otherwise red-flowered, hummingbird-pollinated species were strongly preferred by naïve bumblebees in experimental arrays, and our results suggest this preference reflects multiple floral traits acting in concert to influence pollinator foraging decisions. We found that while increased preference by bumblebees for yellow morphs may provide the potential for a shift in pollinators, this shift appears to be incomplete, as even preferred phenotypes appear ill-suited to these pollinators. This could suggest that current phenotypes may represent an ongoing adaptive walk between peaks in a fitness landscape. Ultimately, this study reveals that suites of floral traits act in concert to shape the attraction, fit, and preference of pollinators, and while changes in attraction traits of large effect like color may initiate the process of a pollinator shift, additional changes to smaller-effect fit traits must follow to ensure efficient pollen transfer and reproductive isolation. Thus, this study sheds light on the order in which floral traits may evolve in an incipient pollinator shift, their genetic basis, and their impact on the behavior of a novel pollinator, with implications for early stages of pollinator-mediated speciation and our understanding of adaptation in complex traits.

## Methods
### Plant material
The yellow-flowered color morph of *M. verbenaceus* (MVYL, hereafter MvY) was originally collected from a population located at Vassey's Paradise in the Grand Canyon, Arizona, USA (exact coordinates unknown), near the northwestern extent of the species range[45], about 100 miles north of the source population of the red-flowered *M. verbenaceus* (MVBL, hereafter MvR; Supplementary Fig. S1A). MvR was collected along the West Fork Trail in Sedona, AZ, USA (34.9883°N, 111.7485°W). Yellow-flowered morphs of *M. cardinalis* have been characterized at two populations: one at the northern extent of the species range in the Siskiyou Mountains of Jackson County, Oregon, USA (known as SM, hereafter McY), and one from the southern range extent in Cedros Island, Baja California. For this study, we characterize the yellow morph from the northern population (SM, exact coordinates unknown), though we note that previous work with pollinators included the southern Cedros Island population[33,45]. The red-flowered reference line of *M. cardinalis* (CE10, hereafter McR) was collected near South Fork Tuolumne River, Tuolumne County, California, USA (37.817°N, 119.867°W). Approximate collection locations were mapped (Supplementary Fig. S1A) in R v4.2.2[87] using packages: sf v1.0-17[88,89], rnaturalearth v1.0.1[90], and rnaturalearthdata v1.0.0 (a free, public domain map dataset[91]). Plants used in experiments represent reference lines inbred for at least ten generations and were grown from seed at the John Innes Centre, Norwich, UK.

Prior to germinating, seeds were sterilized in dilute (1.2%) sodium hypochlorite solution for 5–10 min, then rinsed three times with sterilized distilled water. Seeds were then sown on MS growth media in Petri dishes and placed in a controlled environment room (CER) under 14-hour day lengths at 20 °C day/19 °C night. Following germination, seedlings were transferred to cereal mix to establish in CERs before being moved to the glasshouses at the John Innes Centre.

All floral phenotypic traits were measured on freshly opened flowers (i.e. opened that day), including tissue collected for pigment and RNAseq analyses.

### Floral color: reflectance and pigments
Floral color was quantified by measuring reflectance of fresh corolla tissues (collected immediately prior from plants growing in the glasshouse) using a reflectance spectrometer (FLAME-S-UV-VIS-ES Assembly, 200–850 nm, Ocean Insight), with light source from a pulsed xenon lamp (220 Hz, 220–750 nm, Ocean Insight, v. 2.0.8) on four areas of the corolla: the upper petal lobe, the central (bottom) petal lobe, the lower side petal lobe, and the throat of the corolla tube/nectar guide. The reflectance spectrometer collected readings every 5 ms and averaged 25 scans per reading (Ocean View spectroscopy software, Ocean Insight). Readings were standardized with an absolute white color reference (Certified Reflectance Standard, Labsphere) and electric dark was used as the black standard. Three readings (technical replicates) were taken from each tissue of each flower measured, and three flowers (biological replicates) were measured per individual plant, with 5-16 plants characterized per line (detailed sample sizes in Supplementary Table S1). Reflectance curves at wavelengths from 300 to 700 nm were visualized for all four tissues. All three petal lobe tissues showed similar reflectance curves (Supplementary Fig. S2), so we present and analyze only the central petal lobe tissue hereafter, as this petal lobe is forward facing (not strongly reflexed) and thus visible to pollinators approaching the front of the flowers.

Floral color (reflectance) of the central petal lobe was compared among lines using a principal component analysis (PCA) using R v.4.2.2[87], package psych version - 2.2.5[92] to allow for the varimax rotation, which prioritizes loading each trait (i.e., wavelength) onto only one PC axis, based on the intensity of reflectance at a given wavelength (based on intervals of every ten nanometers from 300 to 700 nm, which captures wavelengths relevant to the vision of most pollinators). First, a PCA was run based on intensity values at all 41 wavelengths; then, only factors with an eigenvalue>1 were used in the PCA, which resulted in a final PCA of 3 factors. To visualize whether these flowers reflect in the ultraviolet (UV) range (including potential patterning), which is important for bee vision, we photographed flowers of 1–3 individuals of all four color morphs under UV light using a UV-sensitive camera (Nikon D610, converted to a Full Spectrum camera by removing the Kolari Vision UV bandpass filter, with a Micro-Nikkor 105 mm lens) in a dark room illuminated by a UV black light.

### Characterizing floral pigments
To characterize the biochemical basis of floral color differences, we analyzed the anthocyanin and carotenoid pigments present in fully opened corolla tissues of five individuals (1 flower per individual plant per pigment group) for all four lines. Total anthocyanins were quantified by measuring absorbance at 525 nm and total carotenoids at 450 nm, using a spectrophotometer (DS-11 FX UV-Vis-Spectrophotometer, Cambridge Bioscience, UK). When initial absorbance readings were above the saturation point of the spectrophotometer, samples were diluted with the addition of extraction buffer (1:1 of 1% solution of hydrochloric acid in methanol for anthocyanins and 4:1 ethyl acetate for carotenoids). Because the size of corollas varied among floral morphs, and thus likely contributed different amounts of tissue, we standardized the absorbance values relative to corolla size by dividing absorbance by the average mass (g) of a fresh corolla for each line (based on an average mass from one corolla from five individual plants), and used these values for analysis (though we note that

overall patterns were similar to those using un-standardized absorbance values). Differences among lines (nested in species) in total anthocyanins and total carotenoids were assessed by performing separate ANOVAs (aov() function in R) for each pigment type, followed by pairwise comparisons based on 95% confidence interval of least squares means using package lsmeans v. 2.30-0 (lsmeans() function, following the Tukey method for $p$-value adjustment for multiple comparisons[93]).

### Anthocyanins

Modified from ref. 94. Roughly, fully opened whole corollas were snap-frozen in liquid nitrogen and ground to a fine powder, anthocyanins were extracted using a solution of 80% methanol and hydrochloric acid with samples left to shake overnight. The resulting supernatant was stored in amber vials prior to analysis. The samples were run on an Agilent 1290 Infinity II UHPLC equipped with PDA and 6546 Q-ToF. Separation was on a 100 × 2.1 mm 2.6μ Kinetex EVO column using the following gradient of acetonitrile (solvent B) versus 1% formic acid in water (solvent A), run at 600 μL.min-1 and 40 °C: 0 min, 2% B; 7.5 min, 30% B; 10 min, 90% B; 10.7 min, 90% B; 10.9 min, 2% B; 15 min, 2% B. The diode array detector collected individual channels at 350 nm (bw 4 nm) and 525 nm (bw 50 nm), and also full spectra from 220–640 nm, at 10 Hz.The Q-ToF collected positive electrospray MS using the Jet-Stream source, together with data-dependent MS2. The MS was calibrated before use, but also had two lock-masses infused during the runs, at 121.05087 and 922.009798; components of Agilent's normal ESI calibration mix. Spray chamber conditions were 325 °C gas, 10 L.min$^{-1}$ drying gas, 20 psi nebulizer pressure, 400 °C sheath gas, 12 L.min$^{-1}$ sheath gas, 4000 V Vcap (spray voltage) with a nozzle voltage of 1000 V. The fragmentor voltage was 180 V.

### Carotenoids

Modified from[95]. Roughly, fully opened whole corollas were snap frozen in liquid nitrogen and ground to a fine powder then carotenoids were extracted through sequential agitation and centrifugation with the following: NaCl saturated aqueous solution, hexane, dichloromethane and ethyl acetate. The final extraction product was stored in amber vials and run on LC/MS. To assist in developing a suitable LC/MS method the following paper was used[96]. The samples were run on an Agilent 1290 Infinity II UHPLC equipped with PDA and 6546 Q-ToF using APCI. The column used was an Agilent Zorbax Eclipse plus C18, 2.1 mm X 50 mm. 1.8 micron, Rapid Resolution HD with an Acetonitrile:Methanol mobile phase at a ratio of 70:30. Flow rate was 0.4 ml/min for 15 min with a column temperature of 32 °C. Violaxanthin, neoxanthin, zeaxanthin, and antheraxanthin were confirmed with standards. For mimulaxanthin, quantification was performed using the response of antheraxanthin.

### Transcriptome analysis and comparative gene expression

RNA was extracted from fully opened whole corollas – including anthers – as was performed for the pigment analyses, in order for these datasets to be comparable. RNA extraction was performed using the SPECTRUM(TM) PLANT TOTAL RNA KIT from Merck Catalogue #STRN50-1KT and eluted in water. Three replicates per genotype were sent for sequencing. Library preparation and sequencing was performed by Novogene using Illumina sequencing technologies. Expression was quantified using kallisto version 0.46.1[97], with the publicly available cDNA dataset from *Mimulus verbenaceus* (MvR = MVBL) reference (http://mimubase.org/). Differential expression data were visualised in Degust[98] with the following parameters; Min Gene CPM = 10 reads in 3 samples, FDR cut-off = 1, abs logFC = 1).

### Whole genome analysis

DNA was extracted from leaf tissue of MvYL and SM using the NUCLEON PHYTOPURE 50 PREPX0.1G Kit (Supplier Catalogue ID: RPN8510) with

the addition of an RNAse treatment using ThermoScientific RNAase T1 Catalogue #EN0541. Library preparation and sequencing was performed by Novogene using Illumina sequencing technologies. All genome and transcriptome data are available on ENA, (accession number PRJEB75514).

Reads were aligned to the publicly available *Mimulus verbenaceus* (MvR=MVBL) and *Mimulus cardinalis* (McR = CE10) reference genomes (http://mimubase.org/) using bwa version 0.7.17[99] with subsequent processing using samtools version 1.7[100] and bcftools version 1.10.2[101].

The analysis of genomic differences described in Fig. 4 were a result of variant calling using bcftools against each line's respective red counterpart, following division of the assemblies into 50 kb windows using the bedtools makewindows function. This produced a table of variants per window, per chromosome (Supplementary Data 1). Figure 4 depicts this variant count with 2nd order smoothing (50 neighbors).

The ITS sequences AY575418.1, AY575414.1, AY575439.1 were downloaded from NCBI, and blasted against the reference sequences of *Mimulus verbenaceus* (MvR = MVBL) and *Mimulus cardinalis* (McR = CE10) reference genomes (http://mimubase.org/). The blast hits of ITS sequences that were full length and in the chromosomes were used to extract the fasta sequence from the reference files using samtools faidx command. See Supplementary Data 2 for fasta sequences. We used Mafft (version 7.520) with default parameters to align the fasta sequences[102]. The tree was built using fasttree (version 2.1.9) with default parameters[103]. The tree was visualized using Interactive Tree Of Life (iTOL)[104].

### Floral scent

We characterized floral volatile organic compounds (VOCs) for 3 technical replicates each of 5–15 plants per line (detailed sample sizes in Supplementary Table S1), from plants growing in the glasshouse at the John Innes Centre. Floral VOCs were collected using dynamic headspace collection as follows. Pairs of freshly cut fully open flowers were placed in 50 mL glass beakers containing approximately 40 mL of sterile ddH$_2$O, which were placed in oven roasting bags (Sainsbury's, UK), sealed and connected to scent traps with aluminum twist ties. Scent traps were composed of modified glass Pasteur pipettes filled with 100 mg Porapak Q 80/100 mesh (Merck/Sigma, #20331), contained within pea-sized amounts of silanized glass wool (Merck/Sigma #20411) on either side, and were connected via plastic tubing to Spectrex PAS-500 volatile pumps (Merck/Sigma, #PAS-500), which were run at a flow rate of 100 mL/min for 24 hours to capture any circadian variation in emissions. Floral VOCs were then eluted from scent traps using 600 uL of extraction solvent (10% acetone in HPLC-grade hexane) into 2 mL screw-top amber glass vials, which were stored at −20 °C in a dedicated scent freezer before concentration and GCMS analysis.

### Compound identification, quantification, and analysis

Sample aliquots of 150ul were first concentrated to 50 ul prior to injection of 3ul into an Agilent GC-MS system (7890B GC with 5977A/B MS) using a Gerstel MPS autosampler system with a splitless inlet held at 250C. The oven temperature was held at 50C for 4 min, then raised at 5C/min to 230C, where it was held for 4 min. The column was a Phenomenex 7HG-6015-02-GGA (35 m × 250 um × 0.1 um) and the carrier gas was helium at a flow rate of 1.0603 ml/min. The MS was run in scan mode for ion masses between 50 and 600, with the MS source held at 230C.

Data files were first analyzed using Agilent Unknowns software (v.10.1) to integrate peaks using the default integrator settings and give the top three tentative NIST library identifications, then processed further via custom shell and Perl scripts. We first excluded peaks with an area under the curve of $1 \times 10^5$ due to the extremely sensitive nature of the default Unknowns integrator, as well as excluding peaks that

eluted after 30 min, which are less likely to be volatile pollinator attractants due to their higher molecular weight. Peaks with the same tentative library identification as a peak within 0.1 min in the blank sample and with less than a 5-fold higher area in the sample of interest were excluded, as were peaks where all three library hits contained silica or phthalate contaminants. Following this, remaining uncertain peaks were compared to the nearest blank peaks (within 0.1 min) by visual comparison of spectra and retained or discarded as above. To further verify the identity of retained peaks, an alkane ladder (Merck, 49452-U) was run at the same time as the samples and the Kovats Retention Index of each peak was calculated and compared with published Kovats indices available via NIST. Compounds with an available authentic reference standard available for purchase (the majority of compounds in this study) were then verified by injection on our GCMS system using the method above and floral sample integrated areas converted to ng/flower/hour using a concentration curve run with each standard. Compounds without a standard available, but with matching Kovats Retention Indices to the literature, had the areas converted using a similarly structured reference compound. Compounds with no matching standard or Kovats Retention Index are listed in Tables 2 and 3 and Table S13 as Unknowns and quantified using a similarly structured reference compound to the best estimate of the NIST library. Scent data were visualized in R v.4.2.2[87] using the vegan package v.2.6-4[105] for NMDS plotting and Welch's $t$ tests for comparisons between morphs within species for both individual volatile and total volatile emissions.

## Floral morphology and nectar

Approximately three fully open flowers each of 5–15 individuals of each line were measured using 150 mm digital calipers (Linear Tools, RS Components) for each morphological floral trait (detailed sample sizes in Supplementary Table S1). Measured traits included corolla tube length (measured from base of corolla to mouth opening), floral display height (vertical spread of petal lobes as viewed from the front), display width (horizontal spread of petal lobes), tube opening height (vertical space within mouth of corolla tube), and opening width (horizontal space within mouth); these traits play important roles in pollinator attraction/visual signals and access to nectar rewards. Herkogamy, the distance between the anthers and stigma of an individual flower, was assessed by measuring the length of the stamens (from the receptacle to the anthers) and the length of the pistil (from the receptacle to the stigma) of three flowers from five individual plants of each line. Herkogamy was calculated as the difference between the length of the pistil and the length of the anthers (allowing for a possible negative herkogamy value if stamens were longer than the pistil).

For nectar measurement, at least three flowers per plant (5–14 plants per line) were sampled on the first day of opening, to ensure consistent time for nectar production across lines. Flowers were collected from plants growing in the glasshouse and immediately sampled for nectar volume and sugar concentration. Nectar was collected from the base of corollas into microcapillary tubes (Cat. No. 2930210, 1.55 mm external diameter, Paul Marienfeld GmbH & Co. KG, Germany), and the height of nectar in tubes was measured in mm using digital calipers, which was then converted into uL (based on capillary tube dimensions (1.15 mm internal diameter and 100 mm length, via manufacturer's specifications) and the geometric formula for volume of a cylinder [height of nectar in mm/100*103.87 = h/100*(π*0.5752)]. Concentration of sugar was then measured using a nectar refractometer (0–50% (BRIX) sugar, Bellingham and Stanley UK Ltd., #12393529).

Multivariate differences in morphological traits (corolla length, display height and width, and opening height and width) were assessed among lines using a nonmetric multidimensional scaling (NMDS) ordination using Bray distances in R v.4.2.2[87] package vegan[105], followed by a MANOVA of these traits by species and line nested within species. Differences among lines for each floral trait were assessed using a linear mixed model (the lmer() function in R v.4.2.2[87] package lme4[106] with line nested within species and individual plant included as a random effect, followed by Type II Wald chi-square tests using the Anova() function in R v.4.2.2[87] package car version 3.0-10[107]. Pairwise comparisons between lines were based on 95% confidence interval of least squares means using package lsmeans v.2.30-0 (lsmeans() function, following the Tukey method for $p$-value adjustment for multiple comparisons[93])

## Floral trait integration analysis

Measured floral traits were divided into four categories: volatiles, color (including both individual pigments and reflectance principal components), morphology, and nectar. Within each category, all pairwise combination Pearson correlation coefficients (PCCs) were calculated between each individual trait pair. Absolute PCC values were compared within each category across lines using Type II Wald chi-square tests using the Anova() function in R v.4.2.2[87] package car v.3.1.3 (107F) with Tukey's Honestly Significant Difference tests applied for volatiles, the only category where the Anova result was significant. No statistical test was performed for nectar, where only one PCC was calculated per line from the two measured traits. Overall correlations between total terpenoid emission and total carotenoid production were similarly analyzed via calculation of the PCC, with species grouped by color (red versus yellow).

For all-trait correlational analyses, Pearson correlation coefficients were calculated for all individual floral traits against one another using the rcorr() function in R v.4.2.2[87] package Hmisc v5.2.1[108]. Principal components for reflectance spectrophotometry data were used in lieu of individual wavelength measurements for this analysis.

## Pollinator perception of color and scent

To characterize how floral colors may be perceived by bee pollinators, we plotted the reflectance data of lower central petal lobes on trichromatic models of bee (*Apis mellifera*) visual systems using R v.4.2.2[87] R package pavo version 2.7.1[109], which estimates level of contrast of a color signal against a vegetative (green) background.

To test whether the floral scents of different color morphs elicit different electrophysiological responses in bumblebees, we performed whole-extract electroantennography (EAG). We exposed antennae from nine naïve lab-reared *Bombus terrestris* ssp. *audax* individuals (see below for hive maintenance conditions; one antenna per individual) to floral scent extractions (concentrated down to half volume) of both yellow-flowered morphs (MvY and McY) and their conspecific red-flowered reference lines (MvR and McR respectively) and measured the electrophysiological response to these extracts, in addition to a positive control floral scent compound (0.5 ng/ul phenylacetaldehyde (PAA, Sigma/Merck #W287407-SAMPLE-K), a compound known to elicit a response to *Bombus terrestris*[110], a negative control (a puff of air through a scent cartridge with no VOCs) and the extraction solution in which the floral VOC blends were suspended (10% acetone in hexane).

Using micro-dissection scissors, single antennae were excised from live bees, which were first chilled for 20 min at 4C, and the tip (last 1.5 flagellomeres) of the antenna was removed using a micro-dissection scalpel under a dissecting microscope. The antenna was then placed onto an antennal fork using electrode gel and placed into an MP-15 micromanipulator-electrode holder mounted to a magnetic base plate. The antenna was allowed to equilibrate for 10 min prior to recording commencing. Recording signals were sent to an IDAC-2 and from there to a computer, where they were recorded using GcEad/2014 software. Stimuli were delivered via a CS-55 stimulus controller. Continuous air flowed over the antenna, with pulses of the same air flow strength controlled via a foot pedal. All EAG equipment and gel were supplied by Ockenfels SYNTECH GmbH, Buchenback, Germany.

The order of scent stimulants was randomized (1 round of 6 exposures each, comprised of 3 pulses 0.5 s apart, delivered every thirty seconds), apart from the positive control (PAA) which was delivered in three rounds of 4 exposures and was always the first, fourth, and final stimulant delivered to each antenna sample. This was done because antennae decrease in responsiveness over time, so the positive control signal was used to normalize antennal responses to other stimuli throughout the course of the EAG assay, and the final corrected amplitude value was then scaled to the initial PAA response (following[111]). Additionally, the first response to a novel stimulus (scent) often results in a disproportionately large peak due to solvent blowoff, so the first response to each stimulus was removed as an outlier for each trial. This resulted in a total of 1230 recorded responses to stimuli (PAA: 315 presentations; air: 153, solvent: 153, MvY: 150, MvR: 153, McY: 153, McR: 153) across nine trials using one antenna from each of nine separate bees, which were measured and recorded using the software Gc-Ead 2014 v 1.2.5 (2014-05-03; SYNTECH, https://www.ockenfels-syntech.com/).

We tested whether the scaled and corrected amplitude of the antennal response varied among stimuli using a linear mixed model (lmer()) with trial (individual antenna) included as a random effect, along with a random effect term of stimuli nested within trial (to account for the repeated exposures of each stimuli to a single individual antenna), followed by Type II Wald chi-square test (Anova() function, 107). Pairwise comparisons among stimuli were assessed based on 95% confidence intervals of least squares means, using the function lsmeans() (package lsmeans v. 2.30-0; 93; using the Tukey method for p-value adjustment) with Sattherwaite degrees of freedom and lmer test limit set to 6000 in R v.4.2.2[87] package lmerTest version 0.9-38[112].

To investigate whether other functional groups of novel pollinators may respond to floral signals in these species, identical electroantennography experiments were also conducted with seven virgin female *Manduca sexta* hawkmoths, which were reared on a standard artificial diet (Frontiers Scientific Services, Newark, DE, USA, #F9783B). Phenylacetaldehyde was again used as a control, as it is known to be detectable by *Manduca sexta*[113]. Details of antennal preparation (although no chilling was used prior to antenna removal), stimulus delivery, and stimulus identity and statistical analysis were identical to the *Bombus terrestris ssp. audax* experiments.

## Bumblebee choice experiment

To assess whether bumblebees exhibited a preference for visiting one floral color morph over another, we performed a pairwise choice experiment using naïve *Bombus terrestris* workers. Hives of *Bombus terrestris ssp. audax* were sourced from Agralan, Ltd. (Wiltshire, UK, #Bumblebee New Season Hive, Replacement) and were maintained in a controlled environment room (CER) in the Entomology Department at the John Innes Centre, Norwich, UK, which was kept at 22C during the day and 20C at night with 14 h day lengths. Hives were kept in a polycarbonate cage and were given access to commercial bee pollen (organic raw bee pollen, Sevenhills Wholefoods, Wakefield, UK) and 25% sucrose (Fischer Scientific) solution. The day (ca. 16:00-17:00) preceding each experimental trial, sucrose solution was removed from the cage, in order to motivate workers to forage during the trial the following day. For use in trials, workers were collected from outside the hive and were kept in individual ventilated 50 mL Falcon tubes, kept in darkness until use.

Open, receptive flowers of yellow- and red-flowered reference lines were cut from plants maintained in the glasshouse and were immediately placed into damp floral foam (Cappstan UK) in 15 mL Falcon tubes. For each within-species pairwise comparison, four flowers of the yellow morph and 4 flowers of the red morph were randomly arranged in one of two 2 × 2 grid arrays on the walls of a 65 × 80 × 78 cm polycarbonate flight cage. Comparisons were made only within-species, so each trial consisted of a pairwise choice of either the red or yellow morph of *M. verbenaceus* (MvR versus MvY) or of *M. cardinalis* (McR versus McY) presented in a 50:50 ratio. The background behind the floral arrays was covered in green poster roll paper (Meadow Green, House of Card & Paper, UK) to simulate a vegetative background. The flight cage was placed inside a CER kept at 20C during experiments.

For each trial, one naïve worker bee was placed in the flight cage with the randomized array of flowers and their foraging behavior was recorded. A choice (floral visit) was recorded any time the bee intentionally contacted the corolla of any flower; if the bee entered the throat of the corolla tube (i.e., to attempt to access the nectar at the base of the corolla tube), this visit was also recorded as a probe. The color morph of each choice was recorded, as well as the handling time (how many seconds the bee handled the flower during each visit/choice). Each trial lasted 30 min, after which time the bee was removed from the flight chamber and euthanized. If a bee failed to forage after 20 min in the chamber, it was removed and euthanized. Flowers that came in contact with bees during a trial were replaced before the next trials and the arrangement of floral morphs in each array was re-randomized for each trial. The flight chamber was cleaned with 70% ethanol between trials of different species to remove any lingering floral volatiles.

Trials were performed until each species comparison had ten successful trials (i.e., ten individual bees had made a choice). Preference of bees was assessed by comparing the observed total number of flower choices made for each color morph to the number of choices expected under the null hypothesis of no color preference (=50%) using a Chi-square goodness of fit test (chisq.test() function in R, 87), with a separate test run for each species, with choices pooled across trials. We also tested for difference among color morphs of the first choice made by each bee (i.e., the color of the flower each bee contacted first) in the same manner. In addition, because preference could vary among individual worker bees, we also assessed preference in a second way using generalized linear mixed models (GLMM) using the glmer() function in R v.4.2.2[87] package lme4 version 1.1-27.1[106], which included a random effect for individual bee. These GLMMs used one of the choice metrics (number of visits, number of probes, and total handling time) as the response variable with floral color morph as the predictor, with a Poisson error distribution useful for count data, and separate models were performed for each species comparison.

## Reporting summary

Further information on research design is available in the Nature Portfolio Reporting Summary linked to this article.

## Data availability

Sequence data that support the findings of this study have been deposited with the European Nucleotide Archive with the accession code PRJEB75514. Metabolomic data and reflectance spectrophotometry data have been deposited with figshare (pigment UHPLC-MS data: https://doi.org/10.6084/m9.figshare.28287782.v1, floral scent GC-MS data: https://doi.org/10.6084/m9.figshare.28287506.v1, and reflectance spectrophotometry data: https://doi.org/10.6084/m9.figshare.28287470.v1) and source data (excluding sequencing and metabolomic data) are additionally provided in the Supplementary Information/Source Data file. Unique seed materials used in this study are available on request from the corresponding author. Source data are provided as a Source Data file. Source data are provided with this paper.

## Code availability

Custom Perl scripts developed for this manuscript are publicly available on GitHub (https://github.com/plantpollinator/RedYellowMimulus/) and archived on Zenodo[114].

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

## Acknowledgements

The authors thank Yao-Wu Yuan for providing seeds for all focal lines, as well as original seed sources provided by R. Vickery (McY = SM and MvY = MVYL), P. Beardsley (MvR = MVBL), and H.D. Bradshaw (McR = CE10). We also thank J. Fant, A. Davies, D. Cohen and members of the Fant lab group for thoughtful feedback on the manuscript, and we thank David Seung for use of the spectrophotometer in his lab at the John Innes Centre. We would also like to thank Baldeep Kular for metabolomic support. Finally, we are grateful to the Metabolomics, Horticultural Services, Informatics, and Entomology Platforms at the John Innes Centre for their support. This work was funded by start-up funds from the John Innes Centre to KJRPB, and by the UK Biotechnology and Biological Sciences Research Council Institute Strategic Programmes (Harnessing Biosynthesis for Sustainable Food and Health (HBio) (grant no. BB/X01097X/1) and Building Robustness in Crops (BRiC) (grant no. BB/X01102X/1). K.E.W. is currently supported by the NSF Postdoctoral Research Fellowships in Biology Program under Grant No. DBI 2208984 (to K.E.W.).

## Author contributions

K.E.W., M.N., and K.J.R.P.B. conceived of and executed the study, performed experiments, and generated the data. K.E.W., M.N., K.J.R.P.B., P.P., P.B., and L.H. analyzed the data. K.E.W., M.N., and K.J.R.P.B. wrote the manuscript, with input from PP. All authors approved the final manuscript.

## Competing interests

The authors declare no competing interests.
