## [Transparent Peer Review file · Nature Communications]

Within-species floral evolution reveals convergence in adaptive walks during incipient pollinator shift

Corresponding Author: Dr Kelsey Byers

Version 0:

Reviewer comments:

Reviewer #1

(Remarks to the Author)

In this manuscript, the authors apply an array of approaches to characterize the color, morphology, and scent of two independently derived yellow morphs of typically red species of *Mimulus*. The wide-ranging data presented are solid and interesting, particularly the evidence of convergence in the level of volatile emissions, corolla width, and increased herkogamy. Further, the manuscript is very well written and clearly presented. At the same time, much of it is confirmatory and, overall, I think the study is lacking in the kind of data that would bring it to the next level. For instance, could they use VIGS or transgenics to explore PELAN paralog function in the MvY morph? Alternatively, could they examine patterns of allelic variation at the PELAN locus across populations of the yellow morphs. Are there distinct alleles under selection? Can they cross the two different yellow morphs together to see if they complement one another or not? I am also curious as to how the morphology of these inbred lines compare with that of the wild populations. If they really want to make an argument about an on-going adaptive walk, having some evidence from wild populations seems important.

Reviewer #2

(Remarks to the Author)

I have just read the manuscript titled "Within-species floral evolution reveals convergence in adaptive walks during incipient pollinator shift" written by Wenzell et al. The work focuses on exploring plant evolution associated with color changes driven by an incipient divergence in the pollinators visiting different morphs. I must say that this work represents one of the few and best-documented examples of coevolution between plants and pollinators. Furthermore, the variety of different tools and results included make this work an outstanding example of multidisciplinary. The results are clear and stand out among the existing literature on this topic, as few authors explore from the genetic mechanism to the effect on the phenotypic character, adding the effect on ecological interaction and the evolutionary divergence pattern that results between morphs or species. It will undoubtedly be an article that attracts the attention of all of us working on pollinator-plant coevolution. The results of the work significantly support the conclusions reached. The methodology is innovative and is interestingly combined in the work, generating a solid outcome. The details on the methods are sufficient, hence the length of the work and the amount of supplementary material it includes. The statistical analyses are appropriate, and the work is very well written, elegantly weaving ideas together.

I would like to draw the authors' attention to the fact that it has been a long time since I have reviewed such a comprehensive and excellent work. Although I have not found any weaknesses to highlight and help them with, I would like to emphasize that it has been a pleasure to read their work.

Reviewer #3

(Remarks to the Author)

I had the opportunity of reading and reviewing the manuscript entitled "Within-species floral evolution reveals convergence in adaptive walks during incipient pollinator shift". Here, the authors demonstrate a scenario of convergent evolution of

yellow morphs in two *Mimulus* species and how this floral color transition is accompanied by a shift in other floral traits. The authors integrated different approximations from phenotype and pollinator behaviour to biochemistry and genomics to successfully address this interesting topic. The resulting work of all this effort is excellent. I have some minor comments: A lot of information is presented which makes difficult to follow some parts. I suggest adding more details in the introduction section about the measurements that will be shown later and why they are important. Not clear the total number of individuals used in each experiment and in all the experiment. Are some measurements are taken in the same individuals? For example, were flowers for pigments characterization, floral scents and phenotype collected from the same plant? Are floral morphology and nectar measurements from the same flower? I suggest including a table with the number of flowers and individuals used for each experiment. Were all measurements taken approximately in the same stage of the life cycle? Any change in these measurements across the flowering was noted? 144-145: I suggest moving this explanation to discussion section or repeat it later with more details. Same for 154-154 lines. Did the authors find any correlation between morphological traits and between those and nectar? Shifts in some traits can be explained because they are highly genetically correlated. Are differences in pollen amount between morphs known? This trait could be interesting to measure in future works. Some degrees of freedom are missing. Reference and version of R software is missing. Add the name of instruments used in Anthocyanins and Carotenoids sections in methodology for readers that are not familiar with these tools and parameters. Add some information about reads as quality threshold or filtering steps if needed. Also, add threshold for significant expression (transcriptomic section).

I hope that the authors find my comments as constructive, and I congratulate them for the work.

Version 1:

Reviewer comments:

Reviewer #1

(Remarks to the Author)

I think the authors did a good job responding to the referees and the work is now acceptable for publication.

Reviewer #3

(Remarks to the Author)

Author note for all Reviewers: Please note that line numbers below refer to the manuscript version without tracked changes.

REVIEWER COMMENTS

Reviewer #1 (Remarks to the Author):

In this manuscript, the authors apply an array of approaches to characterize the color, morphology, and scent of two independently derived yellow morphs of typically red species of *Mimulus*. The wide-ranging data presented are solid and interesting, particularly the evidence of convergence in the level of volatile emissions, corolla width, and increased herkogamy. Further, the manuscript is very well written and clearly presented. At the same time, much of it is confirmatory and, overall, I think the study is lacking in the kind of data that would bring it to the next level. For instance, could they use VIGS or transgenics to explore PELAN paralog function in the MvY morph? Alternatively, could they examine patterns of allelic variation at the PELAN locus across populations of the yellow morphs. Are there distinct alleles under selection? Can they cross the two different yellow morphs together to see if they complement one another or not? I am also curious as to how the morphology of these inbred lines compare with that of the wild populations. If they really want to make an argument about an on-going adaptive walk, having some evidence from wild populations seems important.

Author response: Thank you for the kind review and for your suggestions for ways to take the manuscript forward.

*We agree that looking at the function of the PELAN paralogues in *M. verbenaceus* via VIGS or transgenics would be instructive for further research in this field, however it is outside of the scope of this work. As exemplified in the abstract and keywords summarising this work, the focus of the study was not on molecular or functional characterisation of a single gene or gene family, but rather the large- and accompanying smaller-scale transitions associated with an incipient pollinator shift and their subsequent effect on fitness. We believe this study addresses the 3 objectives laid out in lines 107 to 111 and whilst the experiments suggested by reviewer 1 would be interesting and valuable to future work, we feel they are not essential to our current objectives. We hope to investigate this in a future manuscript focused specifically on the PELAN locus and its role in colour transitions in *Mimulus* more broadly, but we feel it is outside the scope of this paper, which focuses primarily on the phenotypes and pollinator responses involved in the red-yellow shift with supporting whole-genome and transcriptome data. Notably, we have not performed molecular experiments or utilised forward/reverse genetic tools in this manuscript, and thus our findings must as a result be based on sequencing data analyses only.*

*Regarding the reviewer's specific experimental suggestions: there is only one (very small) population of yellow *M. verbenaceus* known, so unfortunately we lack allelic variation to study in the depth suggested, although we are in agreement that this approach could be helpful in future studies of *M. cardinalis*. We also agree that future studies into the morphological and genetic variation of natural populations is a valuable direction for future research, and we have added text to the Discussion to address this (see lines 505-508). A complementation test to look at whether *M. verbenaceus* and *M.**

cardinalis are able to rescue one another's red phenotype would indeed be interesting, but we feel this is again outside of the scope of this paper and would not act to further support the research objectives outlined in the paper. However, future work with a more directed molecular focus could certainly include this approach as well. We will definitely keep the reviewer's suggestions in mind when carrying this work forward for future manuscripts with a more central functional genetics focus.

Reviewer #2 (Remarks to the Author):

I have just read the manuscript titled "Within-species floral evolution reveals convergence in adaptive walks during incipient pollinator shift" written by Wenzell et al. The work focuses on exploring plant evolution associated with color changes driven by an incipient divergence in the pollinators visiting different morphs. I must say that this work represents one of the few and best-documented examples of coevolution between plants and pollinators. Furthermore, the variety of different tools and results included make this work an outstanding example of multidisciplinary. The results are clear and stand out among the existing literature on this topic, as few authors explore from the genetic mechanism to the effect on the phenotypic character, adding the effect on ecological interaction and the evolutionary divergence pattern that results between morphs or species. It will undoubtedly be an article that attracts the attention of all of us working on pollinator-plant coevolution. The results of the work significantly support the conclusions reached. The methodology is innovative and is interestingly combined in the work, generating a solid outcome. The details on the methods are sufficient, hence the length of the work and the amount of supplementary material it includes. The statistical analyses are appropriate, and the work is very well written, elegantly weaving ideas together.

I would like to draw the authors' attention to the fact that it has been a long time since I have reviewed such a comprehensive and excellent work. Although I have not found any weaknesses to highlight and help them with, I would like to emphasize that it has been a pleasure to read their work.

Author response: Thank you very much for your very kind comments on our manuscript! We appreciate the time taken for the review and the positive feedback.

Reviewer #3 (Remarks to the Author):

I had the opportunity of reading and reviewing the manuscript entitled "Whitin-species floral evolution reveals convergence in adaptive walks during incipient pollinator shift". Here, the authors demonstrate a scenario of convergent evolution of yellow morphs in two *Mimulus* species and how this floral color transition is accompanied by a shift in other floral traits. The authors integrated different

approximations from phenotype and pollinator behaviour to biochemistry and genomics to successfully address this interesting topic. The resulting work of all this effort is excellent. I have some minor comments:

Author response: Thank you for the kind review and helpful suggestions, and welcome to the world of manuscript peer reviewing! We have addressed the individual comments below each point.

A lot of information is presented which makes difficult to follow some parts. I suggest adding more details in the introduction section about the measurements that will be shown later and why they are important

Author response: Thank you for this suggestion, please see lines 97 and 99-100 where we have added text to the Introduction (highlighted in yellow) "it remains unknown whether these stable floral morphs also differ in other pollinator-relevant floral traits (including floral morphology, nectar, pigments, and floral scent), which may represent additional steps along an adaptive walk between hummingbird and bee pollination syndromes by further adapting these morphs to bee pollination via modifications to floral attraction and mechanical fit traits."

Not clear the total number of individuals used in each experiment and in all the experiment. Are some measurements are taken in the same individuals? For example, were flowers for pigments characterization, floral scents and phenotype collected from the same plant? Are floral morphology and nectar measurements from the same flower? I suggest including a table with the number of flowers and individuals used for each experiment.

Author response: Thank you for suggesting this. We have now included Table S1 which indicates the sample sizes for flower numbers and individual numbers for each experiment. For the vast majority of measurements, samples were taken from the same set of plants for all measurements in each line (i.e. the same plant was sampled for scent, pigments, reflectance, morphology, and nectar). The exception is herkogamy in McR, where a fresh set of plants were grown as this was not measured in the first set; in addition, throat reflectance was measured in only a subset of plants from the first grow-out in McR. As McR is a >10 generation inbred line, we do not expect that this trait should differ between grow-outs, however.

Were all measurements taken approximately in the same stage of the life cycle? Any change in these measurements across the flowering was noted?

Author response: All traits were measured in first-day open flowers (i.e. those that had opened the day of measurement). Due to the destructive nature of sampling, each category of traits (scent, pigment, reflectance, nectar, and morphology) were measured on different individual flowers from slightly different plant maturity levels from fairly young plants (i.e. sampling started immediately after flowering commenced). We have not previously noted differences in traits when samples were taken

from the first open flowers versus later flowering (note that these plants are perennials, so the flowering season under glasshouse conditions is continuous and very long).

144-145: I suggest moving this explanation to discussion section or repeat it later with more details. Same for 154-154 lines.

Author response: Thank you for the feedback. We have moved lines 144-145 to the Discussion (now lines 447-449): “We also note that the UV “bullseye” pattern seen in yellow morphs may make them more visible to bees, whose eyes have receptors sensitive to UV wavelengths (Briscoe and Chittka, 2001).” We have also added the word “simultaneous” to line 420 of the discussion to re-emphasize the point made in line 154 by the reviewer.

Did the authors find any correlation between morphological traits and between those and nectar? Shifts in some traits can be explained because they are highly genetically correlated.

Author response: Very good question, thank you for raising this! We had previously prepared correlation figures, but didn’t include them in the initial submission, so we’re very glad to see they will be useful, per your feedback. We have now included Figures S6-S9 which demonstrate correlations between all traits in both Pearson correlation coefficient and significance values (lower left and upper right corners respectively), and also add text to this effect in the Results (lines 333-340) and Methods (lines 770-773) sections. We note that there are more significant correlations overall in M. verbenaceus than in M. cardinalis, though this may be biased by the increased number of volatiles present in M. verbenaceus.

Are differences in pollen amount between morphs known? This trait could be interesting to measure in future works.

Author response: We did not measure this trait, but thank you for the suggestion – we shall include it in future work.

Some degrees of freedom are missing.

Author response: Thank you for catching this and our apologies. We have now added degrees of freedom where they were missing (e.g. lines 358-363 in the Results, electroantennography section).

Reference and version of R software is missing.

Author response: Thank you for catching this, again our apologies. We have added this and version numbers for other R packages which had been accidentally left out.

Add the name of instruments used in Anthocyanins and Carotenoids sections in methodology for readers that are not familiar with these tools and parameters.

Author response: Thank you for this suggestion to improve the manuscript. We have added this to the Methods (lines 624-635).

Add some information about reads as quality threshold or filtering steps if needed. Also, add threshold for significant expression (transcriptomic section).

Author response: Thank you, this has now been added in the Methods (lines 656-657).

I hope that the authors find my comments as constructive, and I congratulate them for the work.

Author response: Thank you! We appreciate the effort you put into reviewing the manuscript and your helpful suggestions.